# Low and high frequency intracranial neural signals match in the human associative cortex

Corentin Jacques[1,2†], Jacques Jonas[1,3†], Sophie Colnat-Coulbois[4], Louis Maillard[1,3], Bruno Rossion[1,3]*

[1]Université de Lorraine, CNRS, CRAN, Nancy, France; [2]Psychological Sciences Research Institute (IPSY), Université Catholique de Louvain (UCLouvain), Louvain-la-Neuve, Belgium; [3]Université de Lorraine, CHRU-Nancy, Service de Neurologie, Nancy, France; [4]Université de Lorraine, CHRU-Nancy, Service de Neurochirurgie, Nancy, France

**Abstract** In vivo intracranial recordings of neural activity offer a unique opportunity to understand human brain function. Intracranial electrophysiological (iEEG) activity related to sensory, cognitive or motor events manifests mostly in two types of signals: event-related local field potentials in lower frequency bands (<30 Hz, LF) and broadband activity in the higher end of the frequency spectrum (>30 Hz, High frequency, HF). While most current studies rely exclusively on HF, thought to be more focal and closely related to spiking activity, the relationship between HF and LF signals is unclear, especially in human associative cortex. Here, we provide a large-scale in-depth investigation of the spatial and functional relationship between these 2 signals based on intracranial recordings from 121 individual brains (8000 recording sites). We measure category-selective responses to complex ecologically salient visual stimuli – human faces – across a wide cortical territory in the ventral occipito-temporal cortex (VOTC), with a frequency-tagging method providing high signal-to-noise ratio (SNR) and the same objective quantification of signal and noise for the two frequency ranges. While LF face-selective activity has higher SNR across the VOTC, leading to a larger number of significant electrode contacts especially in the anterior temporal lobe, LF and HF display highly similar spatial, functional, and timing properties. Specifically, and contrary to a widespread assumption, our results point to nearly identical spatial distribution and local spatial extent of LF and HF activity at equal SNR. These observations go a long way towards clarifying the relationship between the two main iEEG signals and reestablish the informative value of LF iEEG to understand human brain function.

*For correspondence:
bruno.rossion@univ-lorraine.fr

†These authors contributed equally to this work

**Competing interest:** The authors declare that no competing interests exist.

## Editor's evaluation

This is an important paper that will be of great interest to researchers interested in neural brain signals at different frequencies. It shows that low-frequency local field potentials and high-frequency (>30 Hz) broadband activity in response to face stimuli have largely similar spatial, functional, and timing properties. The compelling findings are supported by an innovative paradigm and analysis of intracranial recordings in 121 human participants. These observations provide novel basic science insights into how brain responses at different frequencies signal sensory information.

## Introduction

In the last two decades, the direct measure of neural activity from intracranial electrodes implanted in neurosurgical patients for clinical purpose has been increasingly popular among neuroscientists to investigate the neural basis of sensori-motor and cognitive functions. The richness and complexity of the human intracranial recordings approach come partly from the multiplicity of the recorded neural signals. At a macroscopic level of organization, there are two prominent neurophysiological signals. On the one hand, event-related potentials (ERPs; often called local field potentials in iEEG) which are time-locked and largely phase-locked to an event (e.g. a sensory stimulus in the visual modality) and predominant in the lower range of the frequency spectrum (<30 Hz, here referred to as 'low frequency' activity, LF). On the other hand, broadband activity which is largely non phase-locked relative to events and typically observed and quantified over a higher frequency range of the spectrum (>30 Hz also known as 'gamma range'; here 'high-frequency' activity, HF). [Authors' note: LF and HF signals differ both in the frequency range at which they are prevalent, and in terms of their phase-locking relative to the event onset. Event-related responses are (mostly) low-frequency responses while broadband signals are high-frequency mostly non phase-locked responses, distinct from narrow-band gamma oscillations (*Ray and Maunsell, 2011*; *Hermes et al., 2015*). For convenience, in keeping with the frequency range in which they are typically observed, we refer to the event-related/evoked response as 'LF' and the broadband response as 'HF'. Note however that there are cases where phase-locking and frequency range may be dissociated, for instance when using high frequency periodic stimulation (e.g. 15 Hz). In this case, parts of the evoked phase-locked response may fall within the typical range of the high-frequency response, such as the 3rd (45 Hz) or 5th (75 Hz) harmonic of the response e.g. *Winawer et al., 2013*] . A major challenge for the human intracranial approach is to determine which (characteristics) of these signals are most meaningful to understand the neural basis of sensory, cognitive, or motor events. Although early work focused exclusively on characterizing LF activity, the neuroscientific community has now largely shifted its interest to HF signals, which are thought to reflect population-level neuronal firing (*Miller et al., 2007*; *Nir et al., 2007*; *Ray et al., 2008*) and to be more correlated with blood oxygen level dependent (BOLD) activity as recorded with functional magnetic resonance imaging (fMRI) (*Mukamel et al., 2005*; *Hermes et al., 2012*; *Winawer et al., 2013*; *Jacques et al., 2016b*). Compared to low frequency signals, HF also appears to be more straightforward to characterize the time-course of sensory-motor and cognitive processes (i.e. avoiding the issue of varying polarity and morphology of ERP responses), may be more selective to specific stimuli (*Rangarajan et al., 2014*), and is typically assumed to reflect more local neural activity (*Crone et al., 1998*; *Miller et al., 2007*; *Hermes et al., 2012*).

However, due to several factors, the degree of validity of these assumptions, and how they can be generalized across complex human brain functions subtended by large-scale neural networks, is largely unknown. First, in a given study, the simultaneous recording of HF and LF often takes place in a single, relatively small and specific region (e.g. the sensorimotor cortex: *Crone et al., 1998*; *Pfurtscheller et al., 2003*; *Miller et al., 2007*; *Hermes et al., 2012*; the medial occipital cortex: *Winawer et al., 2013*; or the left posterior superior temporal gyrus: *Crone et al., 2001*), and often with limited data sets (4–5 individual brains; N=22 in *Miller et al., 2007*). Second, the two signals are not systematically quantified, with comparisons often limited to determine the number of significant responses for each signal and their overlap (*Crone et al., 1998*; *Crone et al., 2001*; *Pfurtscheller et al., 2003*; *Lachaux et al., 2005*). Third, studies are often limited in their ability to objectively identify, quantify and compare HF and LF signals with the same analysis parameters (*Lachaux et al., 2005*; *Fisch et al., 2009*; *Rangarajan et al., 2014*; *Engell and McCarthy, 2011*; *Vidal et al., 2010*). Altogether, these factors may have led to, or at least substantially enhanced, the typically reported differences between the two types of neurophysiological signals in terms of spatial extent and localization, degree and type of functional selectivity as well as temporal characteristics (*Vidal et al., 2010*; *Winawer et al., 2013*; *Davidesco et al., 2013*; *Privman et al., 2011*; *Nozaradan et al., 2017*).

Here, we provide an original contribution to understand the relationship between stimulus-evoked LF and HF neurophysiological activity in the human brain. Two key aspects of our study allow to circumvent the above-mentioned issues. First, we record the neural system for face recognition, a complex and widely distributed function in the human associative cortex (*Sergent et al., 1992*; *Duchaine and Yovel, 2015*; *Jonas et al., 2016*; *Grill-Spector et al., 2017*), sampled here with a large population (N=121) of implanted individual human brains (>8000 recording sites). Second, we rely on an original

frequency-tagging approach (*Galloway, 1990*; *Norcia et al., 2015*) providing an objective definition and quantification of simultaneously recorded LF and HF activity in the frequency-domain with a high signal-to-noise ratio and the same analysis parameters (*Figure 1*).

Overall, we report (1) highly similar spatial patterns of face-selective LF and HF neural activity including relative local spatial extent; (2) an overall lower face-selective HF amplitude in anterior VOTC regions (i.e. the anterior temporal lobe, ATL), which may explain the underevaluation of this region in recent intracranial studies of higher brain function using HF signal only, and (3) strongly shared functional properties (corresponding amplitudes, face-selectivity, onset time) between LF and HF. While these findings point to largely similar properties between the two types of neural signals, LF category-selective activity has a higher signal-to-noise ratio and allows a more extensive exploration of the ATL, implicated in higher cognitive functions, than HF activity. Altogether, these observations clarify the relationship between two prominent neurophysiological functional activities in the human brain, challenging conventional views and the increasing focus on HF activity in human cognitive neuroscience research.

## Results

Face-selective activity in the VOTC was identified in the frequency domain following a Fourier transform applied either on the raw SEEG signal, highlighting low frequency (LF) activity as in previous studies with this paradigm (e.g. *Jonas et al., 2016*; *Hagen et al., 2020*), or on the time-varying amplitude envelope of the high frequency broadband (HF) signal (*Figure 1D*). Importantly, responses were quantified at the exact frequency of face stimulation (1.2 Hz) and harmonics (*Figure 1E and F*). Significant face-selective responses were determined by grouping of the first four harmonics (i.e. summing 1.2, 2.4, 3.6, and 4.8 Hz, *Figure 2E*) and computing a Z-score transform ($z > 3.1$, $p < 0.001$, *Lochy et al., 2018*; *Jacques et al., 2020*). At this statistical threshold, there were 2130 VOTC recording contacts in the gray matter and medial temporal lobe with significant face-selective activity in 118 participants (among 7374 contacts located in gray matter or medial temporal lobe in the VOTC of 121 participants, that is, 28.8% of all recorded contacts).

### LF dominates HF face-selective activity

Among face-selective recording contacts, 71% showed significant activity only in LF (LF+HF- contacts: 1511/2130; 118 participants, *Figure 2A*, *Figure 2—figure supplement 1*) and 26.7% showed significant activity in both LF and HF signals (LF+HF+ contacts: 569/2130; 101 participants). The remaining 2.3% contacts showed significant activity in HF only (LF-HF+ contacts: 50/2130; 40 participants). This generated a strong asymmetry in contact overlap between signals, where the vast majority (91.9%) of contacts with significant HF activity were also significant in LF, while only 27.3% of contacts with significant LF activity were also significant in HF. The very few LF-HF+ contacts (50 out of 2130 face selective contacts) were spatially scattered with no particular clustering (*Figure 2A*). Similar proportions of contacts were observed using alternative significance thresholds (i.e. different than Z>3.1) (*Figure 2—figure supplement 2*), indicating that the low proportion of LF-HF+ contacts does not stem from the statistical threshold being too severe or liberal. Moreover, there were clear differences between anatomical sub-regions in the proportion of LF+HF+ contacts, which reflect the percentage of overlap between significant LF and HF face-selective responses. For instance, the highest proportion of LF+HF+ contacts across VOTC (disregarding PHG and antPHG in which there were very few contacts) was measured in the right latFG (61.3%), while the lowest was observed in the TP and antMTG/ITG (2.5%, *Figure 2—figure supplement 1*, *Table 1*).

In order to compare LF and HF neural signals, face-selective contacts were grouped in two sets of contacts based on the significance of the response in either LF or HF: (1) LF+ contacts with significant face-selective activity in LF (i.e., LF+HF- and LF+HF+; N=2080 contacts, *Table 1*, *Figure 2A*) and (2) HF+ contacts with significant face-selective activity in HF (i.e., LF-HF+ and LF+HF+; N=619 contacts). Anatomical labeling of each face-selective contact was performed according to the participant's individual anatomy (*Table 1*, *Figure 2—figure supplement 3*) using a topographic parcellation of the VOTC (*Figure 2—figure supplement 3*; as in *Jonas et al., 2016*; *Jacques et al., 2020*). Moreover, to perform group visualization and analyses, the coordinate of each contact was transformed in the Talairach (TAL) space (*Figure 2*).

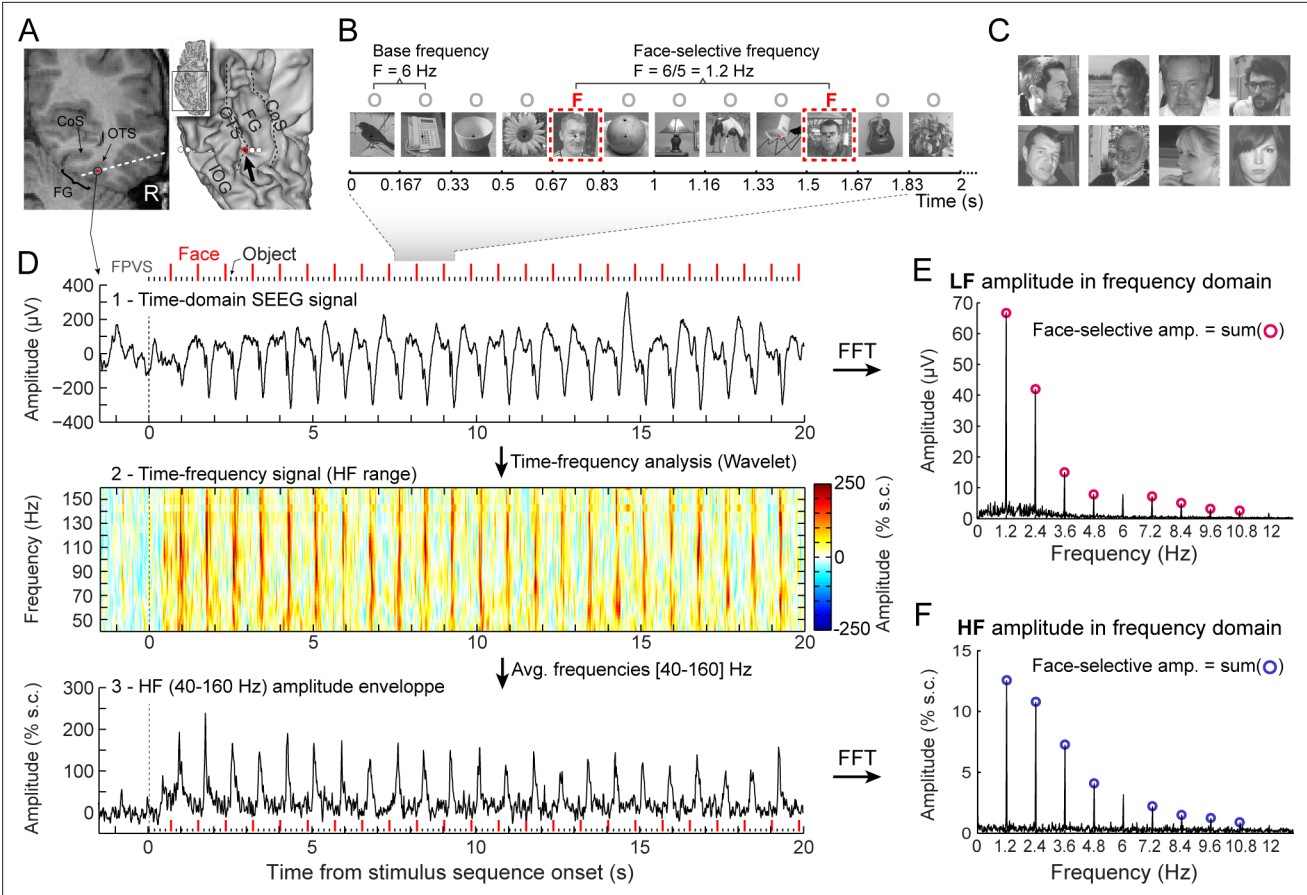

**Figure 1.** Recording and quantifying SEEG low frequency (LF) and high frequency (HF) face-selective signals in the VOTC. (**A**) Left: Coronal slice of an example depth (SEEG) electrode implanted in the right VOTC of an individual participant. Right: the same SEEG electrode array is shown on the reconstructed white matter surface of the participant (ventral view of the right hemisphere). Intracerebral electrode arrays consist of 5–15 contiguous recording contacts (small white rectangles in the coronal slice, white circles on the 3D surface) spread along the electrode length. Electrodes penetrate both gyral and sulcal cortical tissues. Here the electrode extends from the fusiform gyrus to the middle temporal gyrus. The recording contact located at the junction between the lateral fusiform gyrus and occipito-temporal sulcus and where the signal shown in panels D-F is measured is highlighted in red (left: red rectangles surrounded by a circle; right: red circles, see the black arrow). Acronyms: CoS: Collateral sulcus; OTS: Occipito-temporal sulcus; FG: Fusiform gyrus; IOG: Inferior occipital gyrus. (**B**) The fast periodic visual stimulation (FPVS) (or frequency-tagging) paradigm to quantify face-selective neural activity (originally from *Rossion et al., 2015*; see e.g., *Jacques et al., 2016a*; *Rossion et al., 2018*): natural images of nonface objects are presented by sinusoidal contrast modulation at a rate of six stimuli per second (6 Hz) with highly variable face images presented every five stimuli. Common neural activity to faces and nonface objects is expressed at 6 Hz and harmonics in the iEEG signals, while selective (i.e., differential) activity elicited reliably by face stimuli appears at the frequency of 6/5=1.2 Hz. Each stimulation sequence lasts for 70s (2 s showed here). (**C**) Representative examples of natural face images used in the study (actual images not shown for copyright reasons). (**D**) Top: example raw intracranial EEG time-domain signal measured at the recording contact shown in panel A. The signal is shown from –1.5 to 20 s relative to the onset of a stimulation sequence. The time-series displayed is an average of 2 sequences. Above the time-series, red vertical ticks indicate the appearance of face image in the sequence every 0.835 s (i.e. every 5 image at 6 Hz) and small black vertical ticks indicate the appearance of non-face objects every 0.167 s. Example images shown in each sequence are shown in panel B. Middle: a time-frequency representation of the SEEG data in the HF range (40–160 Hz) is obtained with a wavelet transform. The plot shows the percent signal change at each frequency relative to a pre-stimulus baseline period (–1.6s to –0.3s). This highlights distinct periodic burst of HF activity occurring at the frequency of face stimulation (i.e. 1.2 Hz) after the start of the stimulation sequence. Bottom: The modulation of HF amplitude over time (i.e. HF amplitude envelope) is obtained by averaging time-frequency signals across the 40–160 Hz frequency range. Red vertical ticks indicate the appearance of face images in the sequence. (**E**) LF face-selective amplitude is quantified by transforming the time-domain iEEG signal to the frequency domain (Fast Fourier Transform, FFT) and summing amplitudes of the signal at 12 harmonics of the frequency of face stimulation (1.2, 2.4, 3.6, 4.8, … Hz, excluding harmonics of the 6 Hz base stimulation rate). (**F**) HF face-selective amplitude is quantified in the same manner as for LF (panel E) with FFT applied to the HF amplitude envelope.

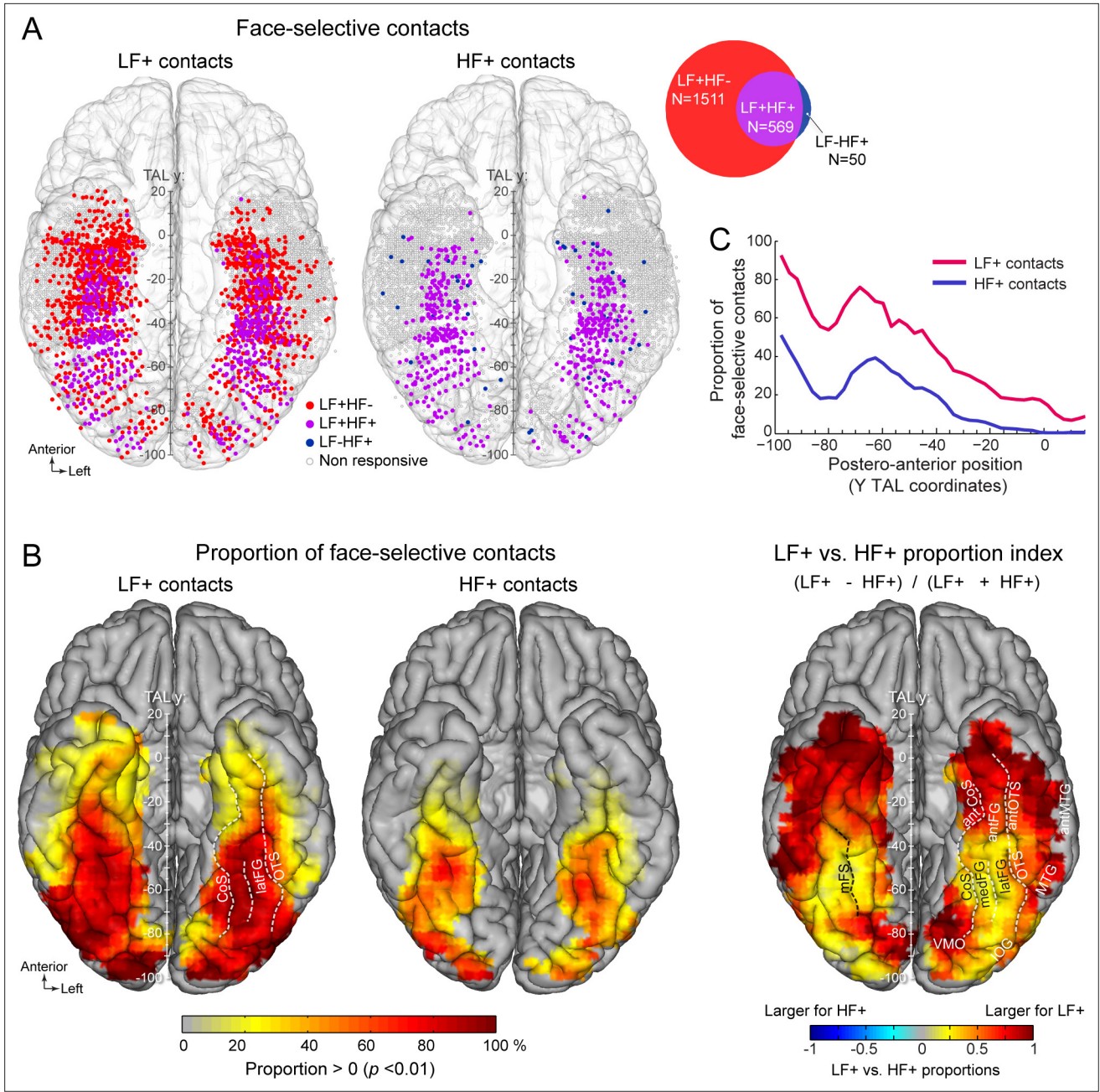

**Figure 2.** Spatial distribution and proportion of LF and HF face-selective SEEG activity over VOTC. (**A**) Map of all VOTC recording contacts across the 121 individual brains displayed in the Talairach space using a transparent reconstructed cortical surface of the Colin27 brain (ventral view). Each circle represents a single recording contact. Each color-filled circle corresponds to a face-selective contact colored as a function of whether LF and/ or HF activity is significant (z-score >3.1, p<00.1) at the contact (for contact count see Venn diagram inset on the right). White-filled circles correspond to contacts on which no significant face-selective activity was recorded. For visualization purposes, individual contacts are displayed larger than their actual size (2 mm in length). Values along the y-axis of the Talairach coordinate system (antero-posterior) are shown near the interhemispheric fissure. (**B**) VOTC maps of the local proportion of contacts showing significant face-selective activity in LF irrespective of HF (LF+, left) and HF irrespective of LF (HF+, middle) relative to the number of recorded contacts, as well as the comparison between the local proportions of LF+ and HF+ contacts across VOTC (right). Proportions are computed using recording contacts contained in 12x12 mm (for x and y Talairach dimensions) voxels. For left and middle maps, only local proportions significantly above zero (p<0.01, percentile bootstrap) are displayed. The map on the right shows an index comparing LF+ to HF+ local proportions computed as the ratio of the proportions of LF+ minus HF+ over the sum of these proportions. Positive values indicate larger proportion of LF+ contacts. (**C**) Proportion of face-selective LF+ and HF+ contacts as a function of the position along the y Talairach axis (postero-anterior) computed by collapsing contacts over both hemispheres. See also *Figure 2—figure supplement 1*, *Figure 2—figure supplement 2*, .

The online version of this article includes the following figure supplement(s) for figure 2:

*Figure 2 continued on next page*

**Figure supplement 1.** Number and proportion of significant face-selective contacts by anatomical region.

**Figure supplement 3.** Labeling face-selective contacts in the individual brain anatomy.

## Similar macro-scale spatial organization of LF and HF signals in VOTC

To compare the spatial organization of LF and HF signals in VOTC, we examined (1) the spatial distribution and proportions of LF+ and HF+ contacts, as well as (2) the spatial patterns of face-selective amplitude in these two types of contacts across the VOTC.

The spatial distribution of face-selective contacts was determined by examining VOTC maps displaying individual contacts in Talairach space, as well as maps depicting the local proportion of face-selective contacts relative to the number of recorded contacts (*Figure 2*). These maps indicate that *LF+ contacts* were widely distributed across the VOTC (*Figure 2*, *Table 1*) with a particular focus in a stretch of cortex going from the IOG, through the FG (particularly in its lateral section –latFG- and adjacent OTS) and up to the antFG and surrounding sulci (antOTS and antCoS). In addition, we observed LF+ contacts in the temporal pole (TP) as well as in subcortical structures of the medial temporal lobe (amygdala –AMG- and hippocampus -HIP). *HF+ contacts* were also distributed across the VOTC with a focus around the same regions as for LF+ contacts (i.e. IOG, FG, antFG, antOTS, antCoS, *Figure 2*). Overall, the proportions of both LF+ and HF+ contacts showed a gradual reduction from posterior to anterior VOTC (*Figure 2C*), and were larger in the right compared to the left

**Table 1.** Number of contacts showing significant responses in LF (LF+) and HFB (HFB+) in each anatomical region.

The corresponding number of participants in which these contacts were found is indicated in parenthesis. For each region, the larger anatomical subdivision is indicated in parenthesis. Acronyms: VMO: ventro-medial occipital cortex; IOG: inferior occipital gyrus; PHG: Parahippocampal Gyrus; medFG: medial fusiform gyrus and collateral sulcus; latFG: lateral FG and occipito-temporal sulcus; MTG/ITG: the inferior and middle temporal gyri; antPHG: anterior PHG; antCoS: anterior collateral sulcus; antOTS: anterior OTS; antFG: anterior FG; antMTG/ITG: anterior MTG and ITG; AMG: amygdala; HIP: hippocampus; TP: temporal pole; OCC: occipital lobe; PTL: posterior temporal lobe; ATL: anterior temporal lobe; MTL: Medial temporal lobe.

| Region | LF+ | | HF+ | |
|---|---|---|---|---|
| | **LH** | **RH** | **LH** | **RH** |
| VMO (OCC) | 104 (16) | 66 (10) | 35 (13) | 23 (9) |
| IOG (OCC) | 65 (16) | 90 (16) | 29 (11) | 49 (14) |
| PHG (PTL) | 5 (3) | 2 (1) | 1 (1) | 0 (0) |
| MedFG (PTL) | 105 (33) | 81 (24) | 64 (28) | 47 (20) |
| LatFG (PTL) | 125 (40) | 105 (31) | 60 (24) | 66 (26) |
| MTG/ITG (PTL) | 44 (25) | 45 (22) | 7 (6) | 5 (2) |
| antPHG (ATL) | 5 (5) | 0 (0) | 0 (0) | 0 (0) |
| antCoS (ATL) | 168 (55) | 148 (52) | 25 (18) | 36 (23) |
| antFG (ATL) | 27 (19) | 27 (17) | 12 (11) | 8 (7) |
| antOTS (ATL) | 198 (66) | 211 (61) | 64 (32) | 58 (34) |
| antMTG/ITG (ATL) | 49 (30) | 114 (45) | 5 (5) | 8 (7) |
| TP (ATL) | 19 (11) | 41 (18) | 2 (2) | 1 (1) |
| AMG (MTL) | 52 (30) | 67 (27) | 6 (5) | 2 (2) |
| HIP (MTL) | 55 (28) | 62 (33) | 2 (2) | 4 (4) |
| Total | 1021 | 1059 | 312 | 307 |

hemisphere (LF+: RH = 31.7%, 1059/3339 vs. LH = 25.3%, 1021/4035, p<0.0001, two-tailed permutation test; HF+: RH = 9.6%, 322/3339 vs. LH = 8.0%, 322/4035, p<0.01). Notably however, across the VOTC, the proportion of LF+ contacts was significantly higher than HF+ contacts (2080/7374=28.1% for LF+ vs. 644/7374=8.7% for HF+, p<0.0001, two-tailed permutation test, *Figure 2B, C*).

Computing a map of the relative decrease of proportion of HF+ relative to LF+ contacts (*Figure 2B*, right) indicates that, while the lower proportion of HF+ contacts was observed throughout the VOTC, it was most evident in the ATL (i.e., anterior to the middle FG). We also found a decrease of proportion of significant contacts in lateral and medial portions of the PTL, apparently producing a more focused distribution of HF+ contacts around the FG region (see below for further analyses of the local spatial extent of both signals).

To further examine the correspondence in the spatial organization of face-selective response across LF and HF signals, we compared the patterns of amplitudes for the two signals across the VOTC. To avoid including noise in the amplitude estimates, LF and HF amplitudes were computed over their respective pools of significant contacts (i.e. LF+ contacts for LF signal and HF+ contacts for HF signal). Amplitude was quantified as the sum of the baseline-subtracted Fourier amplitude over the first 12 harmonics of the face-selective frequency, excluding harmonics of the base stimulation frequency (i.e. 1.2–16.8 Hz excluding 6 and 12 Hz, *Figure 1E, F*). We first examined the winsorized mean amplitudes (see Materials and methods) across face-selective contacts in each individually-defined anatomical region (*Figure 3A*). Overall, the patterns of amplitude variations across VOTC regions are largely similar across the two types of signals. For instance, for both signals, the largest amplitudes were recorded in the latFG, followed by the IOG. This similarity in patterns of amplitude is reflected in the strong correlation between the amplitudes computed in each region and hemispheres (Spearman's Rho = 0.81, 95% C.I.: [0.42–0.99] computed using 17 regions with more than 5 HF+ contacts, *Figure 3B*, *Table 1*). In line with this observation, VOTC maps in TAL space (*Figure 3C*) computed in 12mm x 12mm voxels reveal a striking similarity in the spatial patterns of face-selective response amplitudes for LF and HF signals measured over their respective sets of recording contacts. This similarity manifests in the robust correlation computed between amplitude maps, that is, using mean amplitudes across contacts in each voxel (Pearson' r on log-transformed data = 0.75, [0.71–0.78], *Figure 3D*).

Despite the overall similarity in the spatial organization of the amplitude for LF and HF signals, a notable difference was the stronger right hemispheric advantage for LF compared to HF activity in the latFG. Specifically, while face-selective amplitude was consistently larger in the right compared to left hemisphere in the latFG (right hemispheric advantage: $100*(R-L)/R$=38.7%, effect-size/cohen's d=0.58, *Figure 3A*), this right hemispheric advantage in the latFG was much weaker for HF (right hemispheric advantage = 8.3%; cohen's d=0.09). Statistical tests of interhemispheric amplitude differences for each signal and anatomical regions were performed using linear mixed model statistics. This revealed that for LF responses in LF+ contacts, face-selective amplitude was significantly larger in the right hemisphere in the latFG (p<0.01, FDR corrected), MTG (p<0.01, FDR corrected) and marginally so in the hippocampus (p=0.064; p=0.013 uncorrected; corrected ps >0.28 for other regions). In contrast, no significant interhemispheric difference was found for the HF signal (all corrected ps >0.22; MTG, antFG, antMTG, AMG, HIP not tested due to insufficient number of HF+ contacts in these regions, see *Table 1*).

## Statistical threshold accounts for the low proportion of HF+ contacts in the ATL

In the previous section, we indicate that the proportion of face-selective HF+ relative to LF+ contacts in the ATL appears disproportionately low compared to the posterior section of VOTC (*Figure 2C, D*). This observation could result either from a simple effect of statistical threshold linked to a quantitative difference across signal types and VOTC regions, or from a qualitative difference in the relationship between LF and HF signals in the posterior and anterior VOTC.

To address this issue, we first examined the distribution of face-selective Z-scores (i.e. the value used to determine whether a contact shows a significant face-selective response) across VOTC. We computed Z-scores for all recorded VOTC gray matter contacts (i.e. N=7374 contacts) and visualized the mean Z-score along the postero-anterior axis of VOTC (*Figure 4A*, see *Figure 4—figure supplement 1* for full VOTC maps). This revealed two interesting observations: (1) the mean Z-score is overall

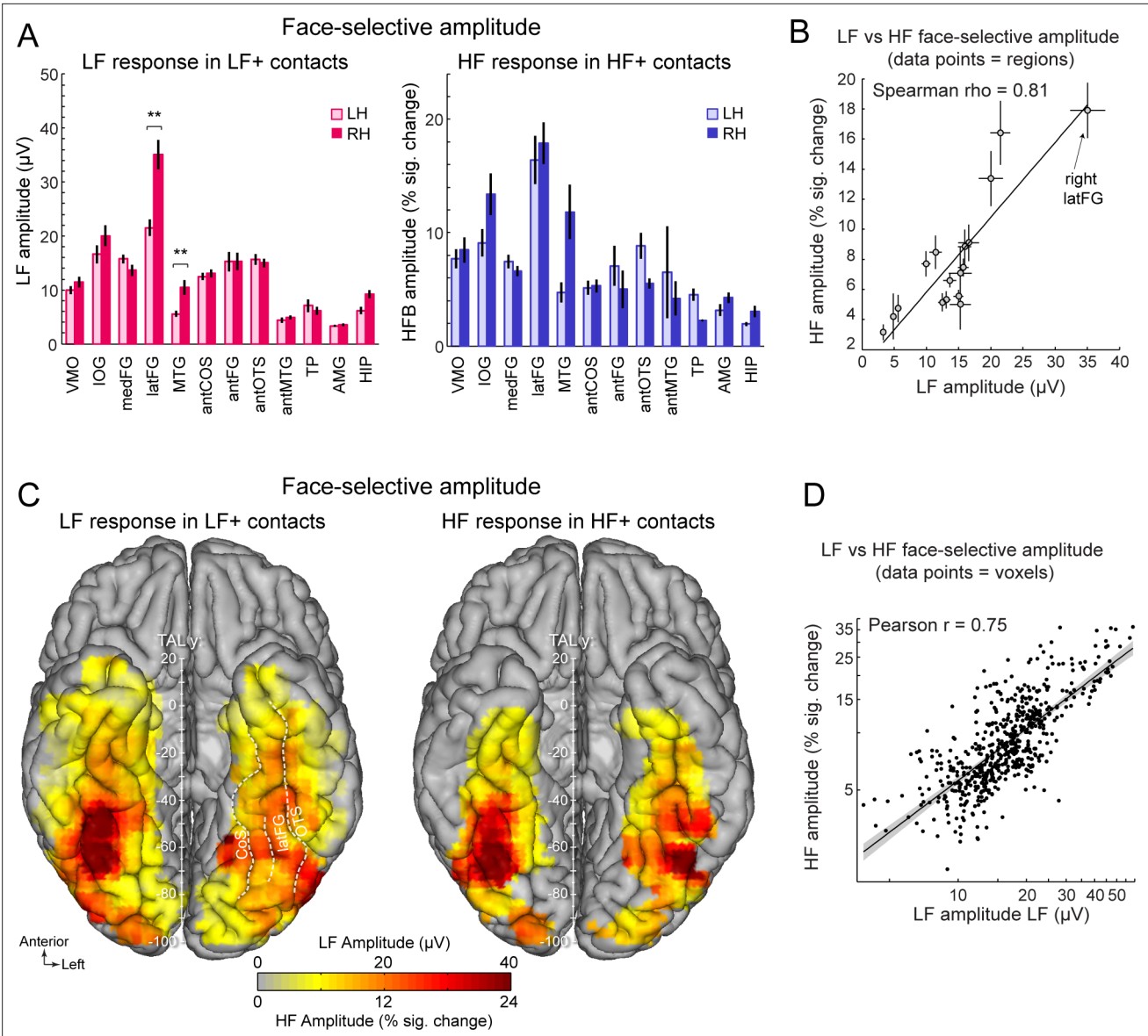

**Figure 3.** Face-selective LF and HF amplitude quantification. (**A**) LF face-selective amplitudes in LF+ contacts (left) and HF amplitude in HF+ contacts shown for each anatomical region (i.e., as defined in the individual native anatomy) and separately for the left and right hemispheres (LH and RH, respectively). Amplitudes are quantified as the mean of the amplitudes across recording contacts within a given anatomical region. Error bars are standard error of the mean across contacts (see *Table 1* for sample size in each region). (**B**) Scatter plot revealing the similarity in the patterns of face-selective LF and HF amplitudes measured in each anatomical region. The amplitude values are the same as in panel A, excluding HIP and TP for which there were too few HF+ contacts. (**C**) Maps showing smoothed LF face-selective amplitude over LF+ contacts (left) and HF amplitude over HF+ contacts (right) displayed over the VOTC cortical surface. Amplitudes are averaged over contacts in 12x12 mm voxels. Only voxels with a proportion of face-selective contact significantly above zero (p<0.01, percentile bootstrap) are displayed. (**D**) Linear relationship between LF and HF amplitude maps shown in panel C. Each data point shows the face-selective amplitude in LF and HF in a 12x12 mm voxels in Talairach space. Amplitudes were normalized using log transformation prior to computing the Pearson correlation. Only voxels overlapping across the two maps are used to estimate the Pearson correlation. The shaded area shows the 95% confidence interval of the linear regression line computed by resampling data points with replacement 1000 times.

higher for LF signal (mean across all VOTC contacts = 3.56 +/- 7.31, *Figure 4A*) compared to HF signal (1.27+/-4.9), and (2) the Z-scores for both LF and HF signals tend to decrease from posterior to anterior VOTC, with an abrupt reduction starting at around TAL coordinate Y = –40 to –30 (*Figure 4A*). These two observations indicate that a significance threshold of Z>3.1 as used here will lead to a lower proportion of significant contacts specifically for HF signal at Y Talairach coordinates more

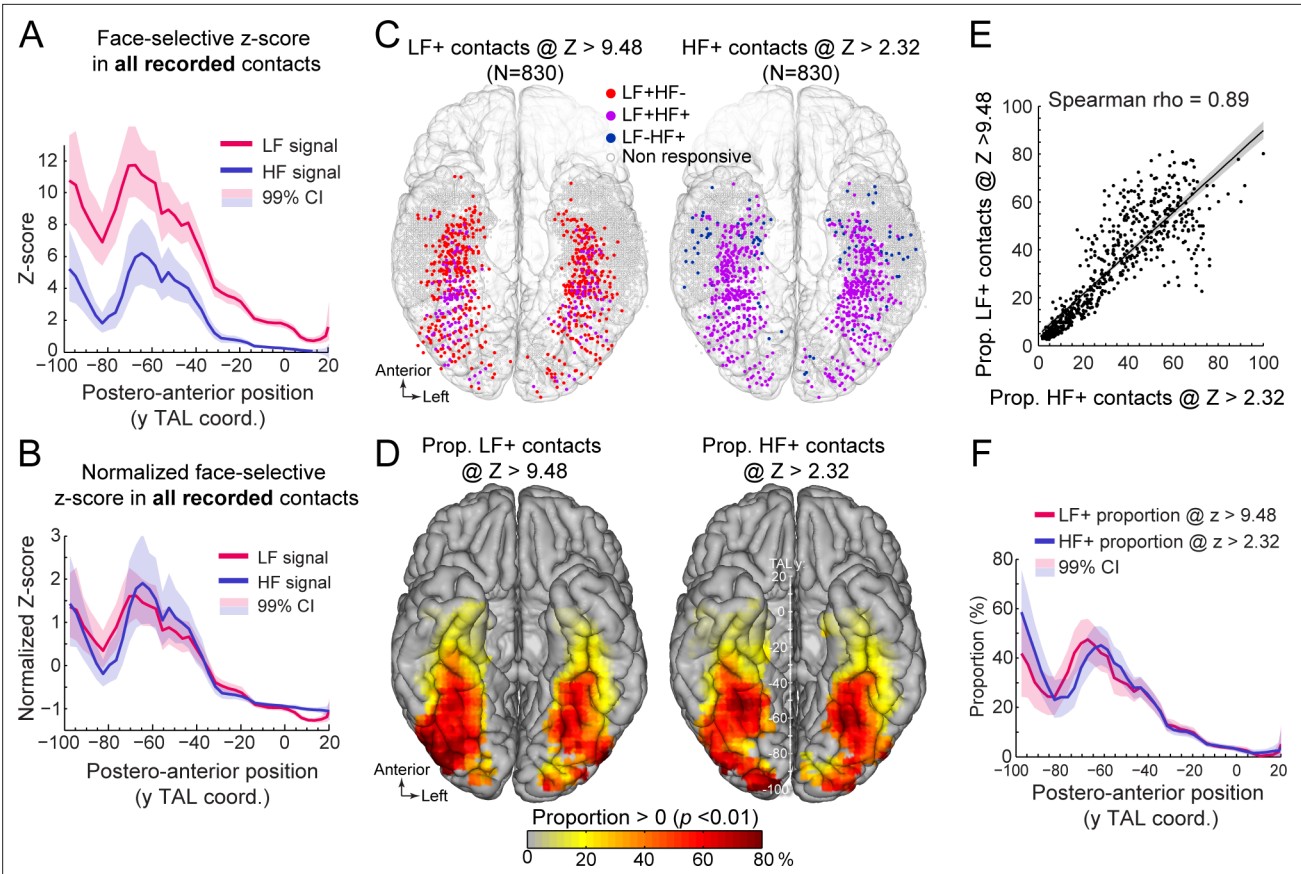

**Figure 4.** Manipulating statistical threshold for LF+ and HF+ contacts. (**A**) Postero-anterior Z-score profiles for LF and HF signals. Z-scores for the face-selective activity measured over all recorded VOTC contacts (i.e. N=7374 contacts) are displayed as a function of the position along the y Talairach axis (postero-anterior; computed by taking the mean Z-score over contacts collapsed across both hemispheres). (**B**) Postero-anterior Z-score profiles (same as in panel A) normalized independently for LF and HF (subtracting the mean and dividing by the standard deviation across postero-anterior positions) to highlight their similarity. (**C**) Spatial distribution of LF+ (left) and HF+ (right) contacts across VOTC after varying the Z-score statistical thresholds (Z>9.48 for LF+ and Z>2.32 for HF+) to equalize the number of recording contacts exhibiting a significant response (i.e. N=830). Color-filled vs. white-filled circle are contacts with vs. without significant face-selective activity at the target Z-score threshold. (**D**) VOTC maps of the local proportion of LF+ (left) and HF+ (right) face-selective contacts detected at two different Z-score thresholds to equalize the number of significant contacts (see panel C). Only local proportions significantly above zero (p<0.01) are displayed. (**E**) Scatter plot displaying the strong similarity between LF+ and HF+ proportion maps shown in panel D. Each data point is the proportion of LF +vs HF+ contacts (i.e., detected using two different Z-score threshold) in a 12x12 mm voxels in Talairach space. Only voxels overlapping across the two maps are used to estimate the correlation. The shaded area shows the 95% confidence interval around the linear regression line (computed by resampling data points with replacement 1000 times). (**F**) Postero-anterior profile of LF+ and HF+ proportions with two different Z-score thresholds (see panels **C**, **D**). See also *Figure 4—figure supplement 1*, *Figure 4—figure supplement 2*.

The online version of this article includes the following figure supplement(s) for figure 4:

**Figure supplement 1.** VOTC maps of face-selective z-scores on all recorded contacts.

**Figure supplement 2.** Manipulating statistical threshold to equate number of LF+ and HF+ contacts (using alternative z-score thresholds compared to *Figure 5*).

**Figure supplement 3.** Exploring the role of signal and noise in variations of Z-score across signals and VOTC regions.

anterior than –30, roughly corresponding to the posterior border of the ATL. Despite the difference in overall Z-score value across HF and LF signals, the postero-anterior profiles of Z-scores for the two signals were extremely similar (Pearson correlation: *r*=0.96). In fact, the Pearson correlation computed *between* LF and HF Z-score profiles was not statistically different from the correlations *within* signals when using split-halves datasets (1000 random split-halves, two-tailed 95% CI for between = [0.9 0.96]; for within = [0.88 0.98]), suggesting that the correlations between LF and HF profiles were at ceiling. This similarity was best visualized after normalizing the Z-score profiles (i.e. by subtracting the

mean and dividing by the standard deviation of each profile, which corresponds to affine transformations of vertical translation and scaling) to minimize distances between profiles (*Figure 4B*).

Second, we investigated how changing the statistical threshold affects the spatial distribution of LF+ and HF+ contacts across VOTC by adjusting the Z-score threshold to exactly match the number of significant LF+ (Z>9.48, n=830 contacts) and HF+ (Z>2.32, n=830 contacts) contacts. Under these conditions, the spatial distribution (*Figure 4C*) and proportion maps (*Figure 4D*) for LF+ and HF+ contacts were extremely similar, with no meaningful statistical difference in local proportion of contacts across VOTC (two-tailed permutation test, fdr-corrected at alpha = 0.05). This was reflected in the high correlation between the two proportion maps (Spearman Rho = 0.890, 95% C.I. [0.86 0.91]; *Figure 4E*). The same was true when statistically comparing the postero-anterior proportion profiles for LF+ and HF+ contacts after adjusting the Z-score thresholds (*Figure 4F*, two-tailed permutation test, fdr-corrected at alpha = 0.05). Pearson correlation confirmed the similarity between the two postero-anterior proportion profiles (*r*=0.94). This lack of meaningful difference is in stark contrast relative to large differences in the postero-anterior profiles that were computed using the same threshold for LF+ and HF+ contacts (*Figure 2C*). Similar observations were made when matching the number of LF+ and HF+ contacts using the original Z>3.1 threshold for HF+ contacts and raising the statistical threshold to Z>12.07 for LF+ contacts to equate the number of HF+ and LF+ contacts (*Figure 4—figure supplement 2*).

To clarify the origin of the SNR difference (computed as the Z-score here) between LF and HF electrophysiological signals, we decomposed the Z-score into its 'signal' and 'noise' constituents. We addressed (1) the lower Z-score for HF compared to LF and (2) the lower Z-score in the ATL relative to posterior VOTC (*Figure 4A*). First, to compare signal and noise between LF and HF signals, we collected these responses over all recorded VOTC contacts and computed a 'signal index' and a 'noise index' (*Figure 4—figure supplement 3*) that account for the 1/f relationship between EEG amplitude and frequency (*Pritchard, 1992*; *Miller et al., 2009*; *Podvalny et al., 2015*). This revealed that the *noise* was similar across the two types of signals (0.28+/-0.03 for LF vs. 0.26+/-0.03 for HF, *Figure 4—figure supplement 3*) but the face-selective *signal* amplitude was on average almost five times larger for LF compared to HF (mean +/-std: 0.83+/-1.34 for LF vs. 0.15+/-0.45 for HF). This indicates that the lower Z-score for HF compared to LF is driven by a smaller face-selective signal, and not by a higher noise for HF. This is also in line with the observation that LF+ contacts with or without significant HF face-selective response differ mostly in the amplitude of the signal (0.40+/-0.87 for LF+HF vs. 9.32+/-9.13 for LF+ HF+ , *Figure 4—figure supplement 3B*) rather than the noise (0.77+/-0.28 for LF+ HF vs. 0.77+/-0.22 for LF+ HF+). Second, we investigated the overall lower Z-score in the ATL (excluding MTL) compared to more posterior VOTC regions (*Figure 4A*). We computed LF and HF signal and noise in three main regions of the VOTC (OCC, PTL, ATL), again using all recorded VOTC contacts (*Figure 4—figure supplement 3*). This revealed that the mean face-selective signal amplitude within the ATL was 72% (LF) and 92% (HF) smaller than in the PTL. In contrast, the noise in the ATL was only 10% larger (LF) or of equal magnitude (HF) than in the PTL. This indicates that the lower Z-scores in ATL are mostly driven by a smaller face-selective signal amplitude in this region compared to the posterior VOTC.

Altogether, these observations indicate that the apparent disproportionate reduction of HF+ face-selective contacts in the ATL region results from a combination of 3 quantitative factors: (1) the face-selective Z-score for the HF signal is overall lower than for the LF signal; (2) the Z-score for both signals decreases along the posterior-anterior axis of VOTC; (3) face-selective LF+ or HF+ contacts are defined using a common statistical threshold. These observations rule out major qualitative differences in the relationship between signals across the VOTC.

## Similar local spatial extent for LF and HF signals

In previous sections, we characterized and compared the spatial distribution of LF and HF signals at a global level across the VOTC. We found a more focused distribution of HF+ contacts around the FG region that could be interpreted as reflecting a narrower spatial distribution of HF responses (*Figure 2B*). To further explore this issue and characterize the spatial properties of LF and HF signals at a finer scale, we took advantage of the high spatial resolution of SEEG recordings (center-to-center distance between contacts is 3.5 mm, whereas it is commonly 10 mm in ECoG studies). To this end, we measured the variation of signal amplitudes along the length of whole SEEG electrode arrays (i.e.

containing 5–15 contacts). To ensure that our analyses included reliable neural responses for both signals in each electrode array, we focused specifically on electrodes arrays containing at least one LF+HF+ contact (N=215 electrode arrays). Electrodes were grouped as a function of the main VOTC region label (i.e. OCC, PTL, ATL) of the recording contact with the maximum amplitude (based on LF signal).

To visualize and compare the spatial extent of LF and HF signals, we investigated the amplitude profiles of the two signals along each electrode array (*Figure 5A*, *Figure 5—figure supplement 1*). This revealed that despite the much larger number of significant contacts for LF compared to HF at the threshold of Z>3.1 (*Figure 2A*, *Table 1*), a large proportion of electrode arrays exhibited highly similar local amplitude profiles for LF and HF signals (*Figure 5A and B*; median Pearson correlation coefficient across 215 electrode arrays = 0.69). Yet, there was a wide range of relationships between LF and HF local amplitude profiles (see *Figure 5—figure supplements 1–3* for the display of all 215 electrode arrays), with only a small number of profiles showing little similarity (*Figure 5A*, bottom row, *Figure 5B*). In addition, as shown on *Figure 5C*, for 77% of the electrode arrays, the maximum amplitude in both signals where either measured on the very same contact (48% of electrodes) or in the directly adjacent contacts spaced 3.5 mm apart (29% of electrodes). For the remainder of the electrodes, the location of the maximum amplitude was 7 millimeter (11%) or more (12%) apart.

We quantified the local spatial properties of LF and HF in 2 separate analyses. To ensure that these analyses were performed on the same neural source for LF and HF in each electrode, we only included electrodes in which the recording contacts with the maximal amplitude for LF and HF were located within 3.5 mm of each other (i.e. 77% of the electrodes; similar results were obtained when including 100% of the electrodes).

As a first estimate of the spatial extent, we determined the size of clusters of contiguous significant face-selective contacts separately for each signal and regions. As expected, the size of these clusters was significantly larger for LF than for HF (mean +/-std = 3.83+/-2.66 contacts vs. 2.29+/-1.58 contacts for LF and HF respectively, p<0.0001, two-tailed permutation test; that is, 13.4 mm vs. 8.0 mm for LF and HF), likely owing to the overall higher Z-scores for LF than for HF (*Figure 4A*).

Second, to quantify the spatial extent of LF and HF signals at a finer scale and, most importantly, independently of the Z-score and statistical threshold, we measured the rate of face-selective amplitude decline as a function of the distance from the maximum amplitude in each electrodes array, which were grouped by main region and hemisphere (*Figure 5D*). Separately for LF and HF signals, electrodes were first spatially centered with respect to the contact with the largest amplitude. For each signal and electrode, amplitudes at contacts located on both sides of the maximum - and equidistant from it - were then averaged in order to 'fold' the electrode around the maximum. The spatial extent for LF and HF signals in each region was estimated by fitting an exponential decay function to the mean amplitude profile (goodness of fit - $R^2$ - ranged from 96.9 to 99.7) and locating the distance from the maximum at which the function reached half of its amplitude range. Across regions, the average spatial extent at half-range ranged from 2.3 to 3.0 mm for LF signal and from 1.9 to 3.4 mm for HF signal (*Figure 5D*). The difference in half-range spatial extent between LF and HF was therefore minimal, with LF being slightly but significantly larger in the left ATL (3.0 vs 2.3 mm for LF and HF respectively, p<0.005, two-tailed permutation test, fdr-corrected). For other regions, differences in spatial extent were not significantly different from each other (−0.5 to 0.5 mm, all ps >0.27). At lower amplitude relative to the peak (i.e. 25% of amplitude range), the spatial extent ranged from 4.8 to 6.0 mm for LF and from 3.8 to 6.8 mm for HF. At this lower relative amplitude, as for the half-range extent, the spatial extent was slightly larger for LF in the left ATL (6.0 *vs* 4.5 mm for LF and HF, p<0.005, fdr-corrected) but no significant difference was observed for other regions (−0.93 to 1 mm, all ps >0.25). We obtained almost identical observations when restricting HF signal to the high-gamma range, that is 80–160 Hz (*Figure 5—figure supplement 4*; *Figure 5—figure supplement 5*).

To sum up, our data highlight two main findings: (1) when using a metric independent of statistical threshold (i.e., rate of amplitude decay), LF and HF have very similar spatial extent, except in the ATL where LF signal is slightly wider; (2) this spatial extent is relatively narrow, with the amplitude being reduced to 50% of the peak amplitude within 2–3.5 mm of the peak and to 25% within 4–7 mm.

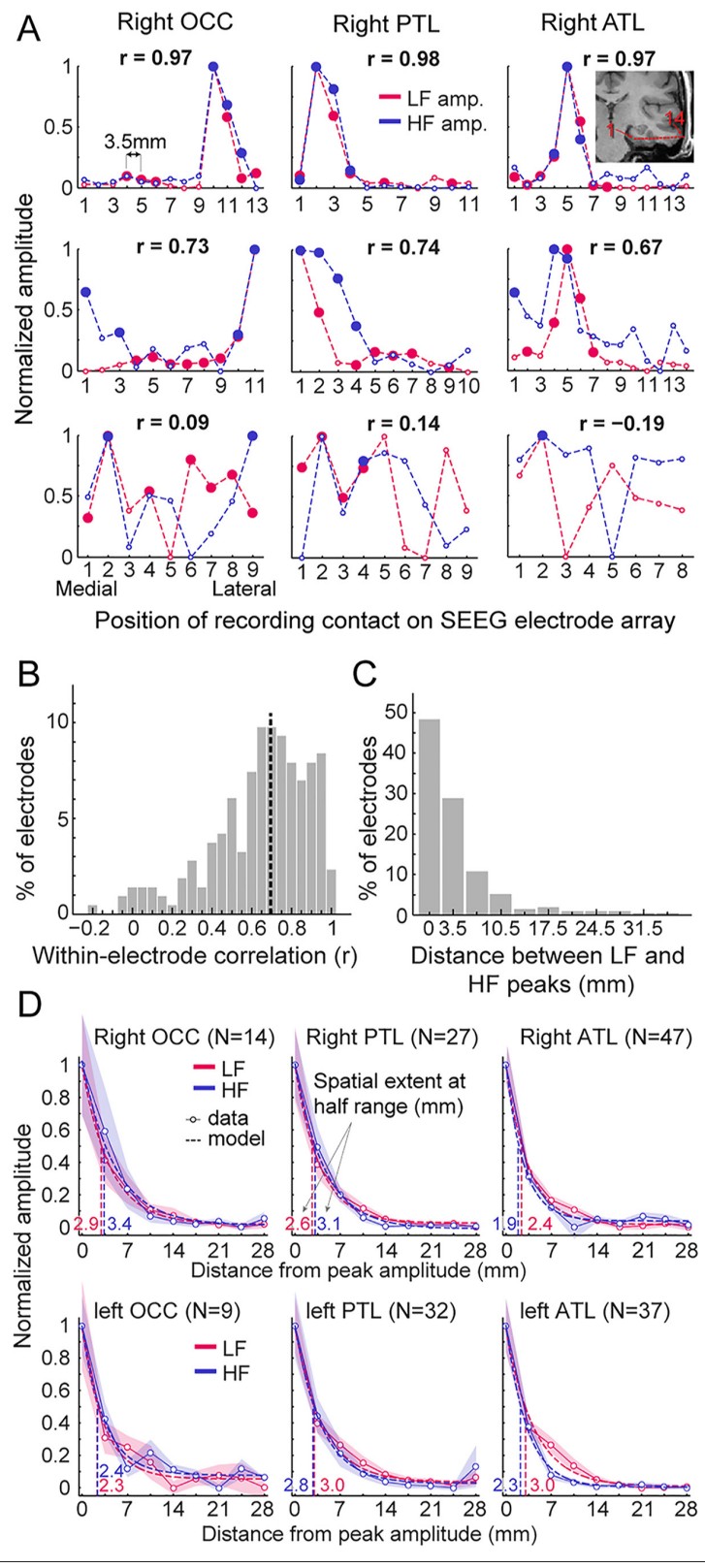

**Figure 5.** Local spatial extent of LF and HF signals. (**A**) Face-selective LF and HF amplitude (normalized between 0 and 1 for display) measured at each contact along a few examples of whole SEEG electrode arrays in the three main right hemisphere VOTC regions (columns). The schematic anatomical trajectory of an ATL electrode is depicted in the inset in the upper right plot. Each plot displays LF and HF signals over the same electrode. Filled

*Figure 5 continued on next page*

*Figure 5 continued*

circles indicate contacts with significant face-selective activity (Z>3.1). Only electrodes containing at least one LF+HF+ contact are included. Plots are vertically ordered by similarity between LF and HF amplitude profiles along the SEEG electrode (quantified using Pearson's correlation): from electrodes with the highest coefficient (top row, see Pearson's r coefficient at the top of each plot), to median (middle row) and worst correlations (bottom row). (**B**) Histogram of Pearson correlations computed between LF and HF amplitudes within each SEEG electrode (N=215 electrodes, see example correlations in panel A). Median correlation is represented by the vertical dashed line. (**C**) Histogram of the distance between LF and HF peak amplitude in any given electrode array. For 48% of all electrodes, the peak amplitude for LF and HF occurred on the same contact (e.g. top row of panel A) and for 29% LF and HF peak amplitudes occur at directly adjacent contacts (e.g. middle row, right column in panel A). (**D**) Local spatial extent of face-selective LF and HF signals in each main VOTC region. Each plot displays the mean variation of face-selective LF and HF amplitude as a function of the distance (mm) from the peak amplitude (located at 0 mm). Only electrodes where LF and HF peak amplitude were at most 3.5 mm from each other were used (see electrode count for each region in parenthesis). Mean amplitudes have been normalized between 0 and 1 for display only; all analyses being performed on non-normalized data. The spatial extent was estimated for each signal and main region by fitting an exponential decay function (dashed lines) to the mean amplitude profile (thin lines) and finding the distance at which the function reach half of its amplitude range. Resulting spatial extents are indicated on each plot and marked by vertical dashed lines. Shaded areas are the standard error of the mean across electrode arrays. See also *Figure 5—figure supplement 1*, *Figure 5—figure supplement 2*, *Figure 5— figure supplement 3*, *Figure 5—figure supplement 4*, *Figure 5—figure supplement 5*.

The online version of this article includes the following figure supplement(s) for figure 5:

**Figure supplement 1.** LF and HF amplitude along SEEG electrode arrays: right OCC and right PTL.

**Figure supplement 2.** LF and HF amplitude along SEEG electrode arrays: right ATL and left OCC.

**Figure supplement 3.** LF and HF amplitude along SEEG electrode arrays: left PTL and left ATL.

**Figure supplement 4.** Scatter plot showing the linear relationship between HF face-selective amplitude measured in the 40–160 Hz frequency range and in the 80–160 Hz frequency range.

**Figure supplement 5.** Comparing LF signal to HF signal measured over the 80–160 Hz range.

## Functional and timing correspondence between LF and HF signals

We further explored the functional relationship between LF and HF category-selective signals by characterizing their respective amplitudes, selectivity and timing.

First, we characterized the functional relationship between LF and HF signals by correlating the face-selective amplitudes across signals using single recording contacts as datapoints. To avoid including noise in correlation estimates, we restricted the analyses to the set of LF+HF+ contacts in which a reliable face-selective activity could be measured for both signals. This revealed a strong relationship when considering responses across the VOTC (Pearson's r on log-transformed amplitudes across 569 contacts: 0.59 [0.53–0.64], p<0.001). Mapping Pearson correlations in Talairach space across the VOTC (*Figure 6A*) revealed that the highest correlations were found in the posterior VOTC, with local peaks in the right latFG and VMO (r~=0.8–0.9), while they were slightly lower in the anterior VOTC. This was confirmed when computing correlations between LF and HF signals separately for each main VOTC region using individual participants' anatomical labels (*Figure 6B*). This revealed similar correlations in the OCC (Pearson's r on log-transformed amplitudes: 0.63 [0.5–0.73]) and PTL (Pearson's r: 0.68 [0.61–0.75]) but a slightly lower LF-HF correlation in the ATL (Pearson's r: 0.45 [0.32–0.58]; OCC vs ATL: p=0.031; PTL vs. ATL: p<0.001, two-tailed percentile bootstrap). Interestingly, within the PTL, the correlation between LF and HF signals was much stronger in the latFG (Pearson's r: 0.73 [0.63–0.80], log-transformed data) than in the medFG (Pearson's r: 0.42 [0.25–0.58]).

The interpretation of the magnitude of a correlation depends on the maximum correlation that can be expected as a function of the noise in the data. Thus, to further characterize the strength of the relationship between LF and HF, we computed an estimation of the maximum expected correlation (MEC) in this dataset (*Figure 6C*), which also relates to test-retest reliability. The noise used to compute the MEC was quantified as the standard deviation in face-selective response amplitude between FPVS sequences. The MEC (averaged across LF and HF) at the level of the whole VOTC was 0.88, 99% CI: [0.85–0.91]. This means that the correlation between LF and HF signals reaches 71%, 99% CI: [67-76]% ( ratio of the actual LF *vs*. HF correlation to the MEC: 0.59/0.83) of the maximum expected correlation in this dataset. For the main regions of the VOTC, the LF *vs*. HF correlations

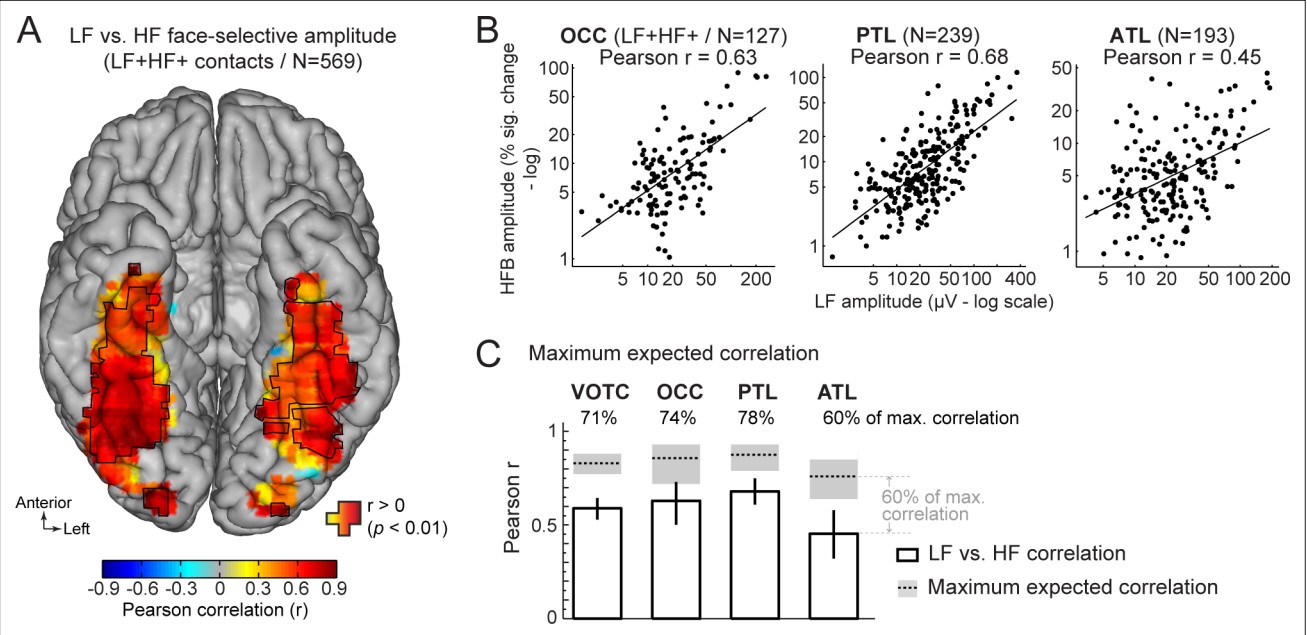

**Figure 6.** Functional relationship between LF and HF face-selective responses. (**A**) VOTC map of Pearson correlations computed between LF and HF face-selective amplitude (log-transformed) measured in LF+HF+ recording contacts. Correlations were computed using contacts located in 15x15 mm voxels. Only voxels containing at least 9 recording contacts are displayed. Significant correlations (p<0.01) are outlined by black contours. (**B**) Scatter plot showing the linear relationship between log-transformed LF and HF face-selective amplitude split by main anatomical region, using LF+HF+ recording contacts as data points. (**C**) Pearson correlation coefficients (white bars) are compared to estimations of the maximum correlation that is expected given the presence of noise in the data (dotted horizontal lines). Error bars and shaded area around the maximum expected correlation (MEC) are 99% confidence intervals. On top of each bar, the ratio of actual correlation to the MEC indicates the percentage of the maximum possible correlation obtained in each region.

reached 74% (OCC: 0.63/0.86; 99% CI: [67-87]%), 78% (PTL: 0.68/0.88; 99% CI: [73-87]%), and 60% (ATL: 0.45/0.76; 99% CI: [54-71]%) of the maximum expected correlation.

Second, we compared the magnitude of face-selectivity across LF and HF signals using a face-selectivity index (FSI) computed by taking the ratio of the face-selective amplitude (i.e. at 1.2 Hz and harmonics) to the sum of the face-selective and general visual (i.e. amplitude at 6, 12, 18 Hz) responses. The FSI varies from 0 (no face-selective response) to 1 (only face-selective response and no general visual response) and allows to quantify the magnitude of the face-selective response relative to the overall visual responsiveness of the cortex around the recording contact. Unlike the face-selective amplitude, which cannot be directly compared across LF and HF, the FSI allows for a direct comparison of selectivity across the two signals. Quantifying the FSI (using LF+HF+ contacts) reveals three important findings. First, the face-selectivity indices were virtually identical between LF and HF signals. Comparing LF to HF FSI for each main VOTC region and hemisphere (**Figure 7A**) revealed no significant difference between signals (range of FSI difference: –0.023–0.041; all ps >0.6, fdr-corrected). Second, selectivity indices increased from posterior VOTC / OCC (LF = 0.64, 99% confidence interval: [0.60–0.69]; HF = 0.62 [0.56–0.67]) to anterior VOTC / ATL (LF = 0.87 [0.85–0.89] and HF = 0.86 [0.84–0.89]) (**Figure 7B**). Third, the face-selectivity indices were larger in the right than in the left hemisphere in OCC (averaging LF and HF: left = 0.57 [0.5–0.64] vs. right = 0.71 [0.65–0.76], p<0.005, fdr-corrected) and PTL (left = 0.74 [0.7–0.77] vs. right = 0.80 [0.76–0.83], p<0.005) regions. In the ATL, the FSI was slightly larger in the left hemisphere (left = 0.89 [0.87–0.91] vs. right = 0.85 [0.82–0.88], p<0.01). Computing a VOTC map of the face-selectivity index in the Talairach space corroborated these three observations (**Figure 7C**). The maps further reveal a low face-selectivity index in medial VOTC for both LF and HF signals, and a high index over more lateral regions, in the strip of cortex from IOG to ATL, where the largest face-selective amplitude is measured. However, unlike face-selective amplitude, the largest face-selectivity index is measured in the ATL regions, owing to the very low response to non-face stimuli (i.e. captured by the 6 Hz general visual response). This means

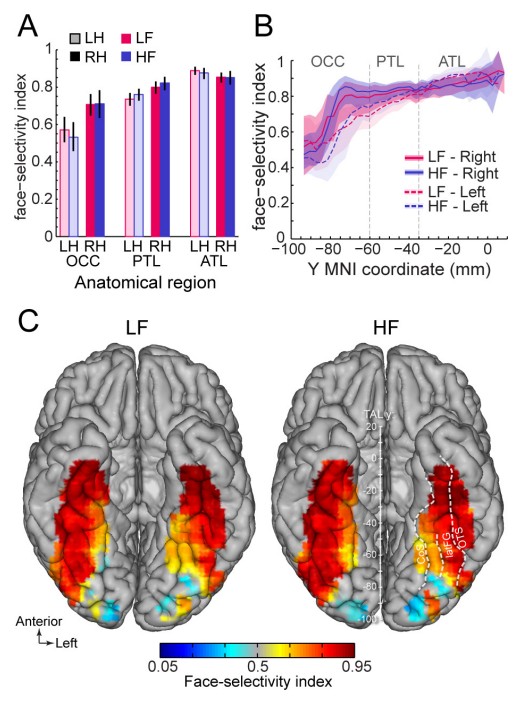

**Figure 7.** Similar face selectivity index for LF and HF. (**A**) Face-selectivity index (FSI) for LF and HF signals computed over LF+HF+ contacts separately for each main region and hemisphere (light color = left hemisphere). Error bars are 99% confidence interval computed using a percentile bootstrap. (**B**) FSI along the antero-posterior axis for LF and HF. FSI is computed in each hemisphere collapsed along the X dimension (medio-lateral). The shaded area shows the 99% confidence interval. Approximate location of subdivision between main VOTC regions in the Talairach space are shown as dashed vertical lines. (**C**) VOTC maps of FSI for LF and HF responses computed over LF+HF+ contacts in 12x12 mm voxels in Talairach space.

that when taking into account the amplitude of the neural response to various visual categories, face-selectivity is similar across the two types of signals.

Third, we explored the time course of LF and HF signals by focusing on the VOTC region with the largest response for both signals: the right latFG. Face-selective time-domain responses were obtained at each contact by selectively filtering-out the SEEG signal generated by the non-face stimuli presented at 6 Hz and harmonics (*Retter and Rossion, 2016*; *Retter et al., 2020*), and segmenting the FPVS time-series around the onset of each face in the FPVS sequences. *Figure 8A* shows normalized LF and HF time-domain face-selective responses in three individual recording contacts with either a high Z-score (left and middle plots), or a middle/low Z-score (right plot). These plots illustrate variations in voltage polarity for LF (mostly positive in the left plot or negative in the middle plot) and variability in SNR across recording contacts (i.e. noisier on the right).

To investigate the correspondence between LF and HF face-selective responses in the latFG we visualized the time-domain response averaged across LF+ HF+ recording contacts (N=65). As shown in *Figure 8B*, both signals started to deviate from baseline at around the same latency (LF: 79ms, 95% confidence interval: [58 - 101] ms and HF: 91ms, [82 - 103] ms), although the LF signal significantly rose above baseline slightly later than HF (102ms, [96 - 108] ms vs. 77ms, [75 - 81] ms for LF and HF respectively) due to higher across-trial standard deviation for LF. In addition, the response duration was longer for LF, returning to baseline level at 639ms (95% confidence interval: [637 641]), compared to HF signal which returned to baseline level at 409 ms [402 - 430].

To further characterize the timing relationship across signals, we estimated the onset latency of LF and HF face-selective response in each right latFG recording contact (rejecting 4 contacts in which we could not reliably determine onset latency). This revealed a highly significant correlation between LF and HF onset latencies (Pearson's *r*=0.48, p<0.001, 99% CI: [0.23 0.67], *Figure 8C*). Despite a significant correlation, the magnitude of the correlation suffers from the uncertainty in the estimated precise onset latency in a number of contacts with lower SNR. Hence, the correlation across latencies was higher when including only contacts with higher Z-score. For instance, Pearson's r was 0.63 when only including the 30 contacts with highest Z-score and peaked at *r*=0.68 for the 16 contacts with highest Z-scores.

## Discussion

Here, we provide a large-scale comparison between two widely used electrophysiological markers of neural activity to investigate human brain function in Neuroscience research: low-frequency signals, that are both time-locked and phase-locked to the stimulation, and high frequency broadband signals, which are time-locked but largely non phase-locked to the stimulation. To this end, we analyzed an unusually large dataset of 121 participants providing more than 8000 recording sites across the

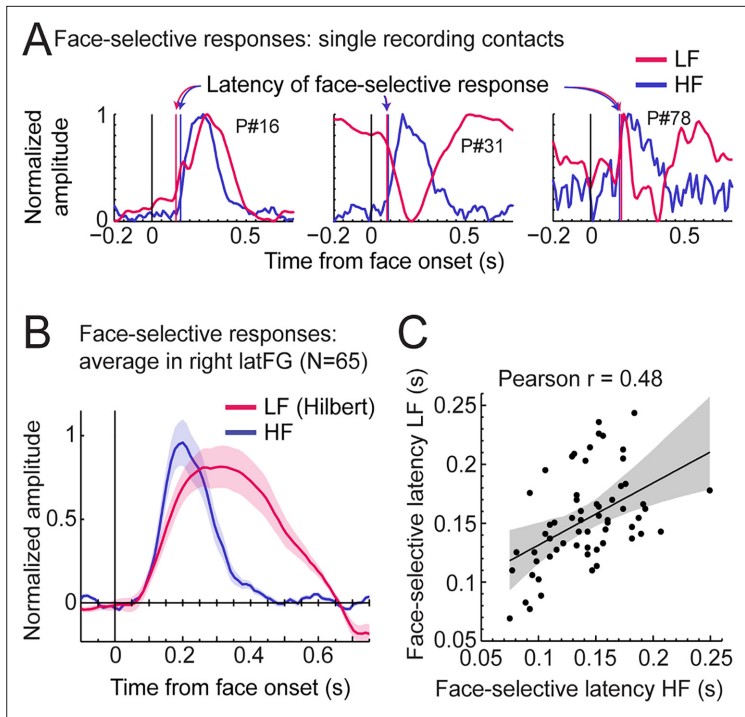

**Figure 8.** LF and HF timing relationship in the right latFG. (**A**) Mean time-domain face-selective responses for LF and HF in three example recording contacts (from three different participants) with high (left, middle) or middle/ low (right) face-selective Z-scores. LF and HF time-domain signals in FPVS sequences were segmented relative to each face onset (i.e. every 0.835 s). The signal related to the general visual response at 6 Hz and harmonics was selectively filtered-out, and resulting segments were averaged. Due to normalization (between 0 and 1 for display purpose), responses of differing polarity across signals result in misaligned pre-face-onset levels (e.g. middle plot, –0.166–0 s). Vertical lines show estimated onset latency of face-selective responses for LF and HF. (**B**) Time-domain face-selective responses averaged across 65 LF+HF+ recording contacts in the right latFG. The shaded area shows the standard error of the mean across contacts. For LF, to limit the influence of variation in response morphology or polarity across recording contacts, a Hilbert transform was applied to the response of each contact before averaging. Averaged time-domain responses were then normalized (0–1) and aligned for their pre-face-onset amplitude level (–0.166–0 s). (**C**) Scatter plot showing the relationship between the onset latency of LF and HF face-selective responses measured in individual recording contacts in the right latFG (see vertical lines in panel A). The shaded area shows the 99% confidence interval of the linear regression line computed by resampling data points with replacement 1000 times.

whole VOTC. We measured neural activity selectively triggered by the presentation of human faces, an optimal stimulus category to probe the ventral pathway for visual object recognition (**DiCarlo et al., 2012**; **Grill-Spector et al., 2017**; **Rossion et al., 2018**). Our original approach combines (1) intracerebral recordings, which give access to both sulci and gyri, and (2) frequency-tagging to maximize homogeneity across the two signal analyses pipelines, signal-to-noise ratio, and objectivity of measurement of both signal and noise (i.e., at pre-defined frequency bins). It reveals three main findings overall.

First, we find highly similar spatial patterns of LF and HF face-selective activity across brain regions, in terms of significant contacts overlap, relative amplitudes and local spatial extent. Second, our analyses show that genuine category-selective neural activity in the anterior portion of the VOTC, the VATL, is likely to be missed with HF signals only. Finally, these two face-selective neural signals are highly functionally related as indicated by their strong amplitude correlation, degree of face-selectivity, and concurrent onset timing.

Overall, these findings point to largely similar functional properties between these two major neural signals, which may, at the current state of knowledge, therefore provide essentially the same type of information regarding the neural basis of human (re)cognition. Yet, questioning the current focus on HF signals at the expense of LF signals in intracranial human recording research, our observations

highlight significant advantages of LF signals: higher SNR to identify a larger number of significant responses, stronger right hemispheric dominance and more extensive ATL exploration. Altogether, these observations, which are discussed more specifically below, re-establish the high value of LF in mapping neural activity in human associative cortex with iEEG.

## Spatially overlapping face-selective LF and HF neural activity

The wide spatial distribution of face-selective LF neural activity across the VOTC has been previously described with smaller samples (*Jonas et al., 2016*; *Hagen et al., 2020*), and is extended here to HF signals isolated with the same experimental approach. To our knowledge, only one human intracranial recording study compared the spatial overlap of HF and LF neural face-selectivity, focusing on the IOG and latFG (*Engell and McCarthy, 2011*) and concluding in favor of spatially and functionally dissociated signals indexing face-selectivity. However, this reported dissociation could be due to substantial methodological differences for sampling LF and HF signals in that study, i.e., raw N200 peak amplitude in the 0.16–0.24 s time-window for LF *vs.* average power in a 0.2–0.6 s time-window for HF [Note also that in *Engell and McCarthy, 2011*, a large percentage of electrodes classified as showing HF-only face-selective responses since they did not meet strict criteria for face-selective N200s nevertheless exhibited a large face-selective component at a later time point (P290). This suggests that using the same time window for quantifying LF and HF signals might have reduced the number of HF-only responses and decreased the spatial dissociation between signals in that study]. In contrast, here with highly similar methods and criteria to quantify signals in the two frequency bands, significant face-selective LF activity was found on the vast majority of face-selective HF+ contacts (i.e. 92.7%), with the remaining very few HF-only responses being spatially scattered and thus likely to reflect noise. In particular, the strongest overlap was found in the regions showing the largest proportion and amplitude of face-selective activity such as the IOG and latFG. Overall, our results point to largely overlapping LF and HF functional neural activity, especially from the point of view of HF, which is almost never found in isolation.

Along these lines, the proportions of LF+ and HF+ contacts throughout the VOTC showed similar local and global patterns: highest proportions in the latFG and IOG, a progressive reduction toward the ATL, and a right hemispheric dominance. Yet, the proportion of LF+ contacts is higher overall and more broadly distributed than for HF, which appears to be in agreement with previous observations (*Engell and McCarthy, 2011*; *Fisch et al., 2009*; *Vidal et al., 2010*). Here we demonstrate that the higher number of LF+ contacts is merely due to the larger SNR for LF as compared to HF signal, so that equating the number of LF+ and HF+ contacts by adapting statistical threshold results in indistinguishable spatial distributions between LF+ and HF+ contacts (*Figure 4*).

Beyond the spatial overlap between LF and HF significant recording contacts, there were also robust spatial correlations between response amplitudes, either across anatomical regions or voxels (*Figure 3*). One noticeable difference though, is the stronger right hemispheric advantage in LF than HF amplitude in the latFG, which is important given the well-known dominance of the right hemisphere in face recognition in the human species, especially in the latFG (*Rossion and Lochy, 2022*, for review).

## HF underestimates face-selective ATL activity

As shown previously with fMRI (*Collins and Olson, 2014*) but mainly with direct recordings of neural activity that do not suffer from magnetic susceptibility artifacts in this region, substantial face-selective neural activity is found in the human (ventral) ATL (*Jonas et al., 2016*; *Hagen et al., 2020*; see also *Allison et al., 1999*). Here, the highest amplitude and proportion of LF+ contacts were found in the antFG and adjacent sulci: the antCoS and antOTS. A smaller proportion of LF+ contacts were found in the temporal pole and in the anterior inferior and middle temporal gyri. In comparison, there was a disproportionately low number of face-selective HF+ contacts in the ATL. In the current study, the large sample of participants and number of recording contacts in the ATL allowed to reveal (a low proportion of) HF+ contacts in this region and to show their spatial and functional correspondence to LF signals. However, with more limited datasets as in most intracranial EEG studies (i.e. usually 2–15 subjects) and a data analysis restricted to HF signal, the ATL would not have been classified as a face-selective region. As a matter of fact, most previous intracranial EEG studies focusing on HF reported no or little face-selective ATL activity (e.g. *Sanada et al., 2021*; *Schrouff et al., 2020*;

*Jacques et al., 2016b*; *Davidesco et al., 2013*; *Miller et al., 2017*; *Norman et al., 2019*; *Rangarajan et al., 2014*; *Kadipasaoglu et al., 2016*). Given that fMRI barely record any face-selective ATL activity (except very anteriorly in the temporal lobe, see *Collins and Olson, 2014*), likely due to signal drop-out (*Ojemann et al., 1997*; *Winawer et al., 2010*), it is not surprising that this region is largely neglected, even in recent reviews on the functional architecture of face perception in the VOTC (*Grill-Spector et al., 2017*). More specifically, there appears to be a substantial spatial 'gap' between the FFA/mFus-faces in the middle fusiform gyrus and rarely disclosed ventral ATL face-selective area(s) close to the temporal pole (for recent illustrations of this functional spatial gap in face-selectivity; see e.g., Figure 2 in *Collins and Olson, 2014*; Figures 1 and 4 in *Wang et al., 2020*; Figure 2 in *Kovács, 2020*; Figure 1 in *Volfart et al., 2022*). This gap corresponds in particular to the location of the antFG and adjacent sulci, that is, the antCoS and the antOTS, in which particularly large LF face-selective activity is found in the current study. Thus, overall, it is fair to say that it is the exploration of LF signal in intracranial electrophysiology that has truly brought forward the ATL as a major face-selective region, especially the antFG and adjacent sulci (*Jonas et al., 2016*; *Hagen et al., 2020*; *Jacques et al., 2020*; *Volfart et al., 2022*; the current study). In order to reveal the full spatial extent of the human cortical face network and investigate the nature of representations and processes in these (ventral) ATL regions, it is critical that future intracranial studies do not limit their investigation to HF neural activity but also fully explore LF signals.

Is this larger proportion of LF+ compared to HF+ contacts in the ATL explained by qualitative or quantitative differences across signals? In other words, is there a modulation of the relationship between LF and HF signals from posterior to anterior VOTC (i.e. a qualitative difference) or is this relationship stable across the VOTC (i.e. a quantitative difference)? Here we showed that this difference is accounted by a combination of three quantitative factors: (1) an overall higher SNR (Z-score) for LF signal, (2) a lower Z-score overall in anterior VOTC, (3) the use of a common statistical threshold for both signals (Z>3.1, p<0.001). Thus, according to these findings, the disproportionately lower proportion of HF+ responses in the ATL results mostly from a quantitative difference between LF and HF. In addition, we showed that the decrease in face-selective *signal* amplitude is much sharper than the increase in *noise* level. These observations indicate that the lower Z-score in the ATL (for both LF and HF but more so for HF) is predominantly caused by a smaller face-selective signal amplitude in this region compared to more posterior regions. Future intracranial human recording studies will have to determine whether the lower face-selective signal amplitude in the ATL relates to physiological properties of the cortex in this region, potentially generating weak population-level neural activity, and if it can be increased by using more suitable stimuli (e.g. familiar faces linked to semantic information; *Collins and Olson, 2014*; *Rice et al., 2018*) for this region.

## Similar local spatial extent for LF and HF when accounting for differences in signal amplitude

In human intracranial recordings, it is generally assumed that HF signals reflect a more local neural activity than LF signals and hence provide higher spatial resolution (*Crone et al., 1998*; *Crone et al., 2006*; *Pfurtscheller et al., 2003*; *Miller et al., 2007*; *Hermes et al., 2012*; *Lachaux et al., 2012*). However, this assumption rests on findings from studies that have quantified the spatial extent by counting the number of recording sites with significant responses for LF and for HF (*Miller et al., 2007*; *Crone et al., 1998*; *Pfurtscheller et al., 2003*). In these studies, LF generate larger clusters of significant responses than HF, which is taken as an indication that the latter type of signal is more focal than the former. Here, we also observed a larger number of LF+ contacts and a larger mean number of contiguous LF+ compared to HF+ contacts. However, one important issue with the current and previous finding is that it may result either from (1) a higher amplitude or SNR at the neural source of the signal (as observed in the current study for LF), generating signals that stay above the significance threshold further away from the actual source (potentially due to simple volume conduction), or from (2) a genuine larger 'size' of the neural source (e.g. its cortical surface). In other words, counting significant responses confounds the quantification of spatial spread or 'reach' of the signal, and the quantification of the spatial extent of the cortical source. To circumvent this issue, in addition to quantifying the number of contiguous significant contacts, here we also quantified the spatial extent by computing the rate of amplitude decay as a function of distance from the peak amplitude, using all recording sites independently of the statistical significance of the signals at these sites. Systematic

differences in the size of the neural source generating LF and HF should result in differences in the rate of the amplitude decay for these signals. This method is similar to what is typically used in the estimation of the spatial extent or magnification factor in lower-level visual cortex in previous studies in macaques or humans (*Dubey and Ray, 2019*; *Xing et al., 2009*; *Yoshor et al., 2007*). Using this approach, and in contrast to the above prevalent view, we found a similar spatial extent for HF and LF signals, estimated to be between 1.8 and 3.5 mm radius. Interestingly, this is only slightly larger than a recent estimate of spatial extent in ECoG recordings in the macaque's V1 cortex (~1.5 mm, *Dubey and Ray, 2019*). Overall, this lack of a difference in the amplitude decay function suggests a comparable cortical area generating the two signals, at least at the scale of our recordings, and a comparable source geometry (i.e., location, orientation and number of sources and sinks in the cortical layers, *Herreras, 2016*). This finding is also in line with our observation that differences between face-selective HF and LF activity are mostly quantitative (i.e. a difference in SNR) rather than qualitative (i.e. a difference in the amplitude decay function).

Several additional factors may explain the discrepancy between our and previous findings from human intracranial recordings in terms of relative spatial extent of HF and LF. First, the view that the HF signal is more focal than the LF signal is partly inspired by the finding that the HF signal measured with micro-electrodes correlates with neuronal spiking activity (*Manning et al., 2009*; *Ray et al., 2008*; *Nir et al., 2007*). Specifically, given the reduced spatial and temporal summation of field potentials for spiking activity compared to lower frequency components of LFPs (*Lindén et al., 2011*) and the resulting more local activity of spiking neurons compared to lower frequency LFPs (e.g. *Gray et al., 1995*; *Buzsáki, 2004*), it is often assumed than HF measured in ECoG or SEEG is more focal than responses in LF (e.g. *Lachaux et al., 2012*; *Crone et al., 2006*). However, spiking activity and HF do not always correlate, especially in superficial layers close to the cortical surface (*Leszczyński et al., 2020*) where macro-electrode HF is typically measured as in the ECoG studies that have highlighted a more focal activity for HF compared to LF (*Miller et al., 2007*; *Crone et al., 1998*; *Pfurtscheller et al., 2003*; *Hermes et al., 2012*). Moreover, the assumption about the spatial relationship between LF and HF in SEEG/ECoG measured through macro-electrodes is based on a direct extrapolation of the relationship between HF and spiking activity highlighted with micro-electrodes. This neglects the vast difference in the size and recording surface between micro- and macro-electrode recordings (1–2 orders of magnitude difference), and thus potential differences in the neurophysiological origins of HF (and thus also the spread of HF propagation in brain tissue) when measured with micro-electrodes or macro-electrodes. In other words, the spatial properties of HF measured with micro-electrodes may not translate to HF signal measured with macro-electrodes.

Second, the hypothesis for a more focal HF compared to LF signals is mostly supported by recordings performed in a single region, the sensorimotor cortex (*Miller et al., 2007*; *Crone et al., 1998*; *Pfurtscheller et al., 2003*; *Hermes et al., 2012*), which largely consists of primary cortices. In contrast, here we recorded across a very large cortical region, the VOTC, composed of many different areas with various cortical geometries and cytoarchitectonic properties. Moreover, by recording higher-order category-selective activity, we measured activity confined to associative areas. Both neuronal density (*Collins et al., 2010*; *Turner et al., 2016*) and myelination (*Bryant and Preuss, 2018*) are substantially lower in associative cortices than in primary cortices in primates, and these factors may thus contribute to the lack of spatial extent difference between HF and LF observed here as compared to previous reports.

Third, previous studies compared the spatial properties of an *increase* (relative to baseline) in HF amplitude to the spatial properties of a *decrease* (i.e. event-related desynchronization) of LF amplitude in the alpha and beta frequency ranges (*Crone et al., 1998*; *Pfurtscheller et al., 2003*; *Miller et al., 2007*; *Hermes et al., 2012*). This comparison may be unwarranted due to likely different mechanisms, brain networks and cortical layers involved in generating neuronal increases and decreases (e.g. input *vs.* modulatory signal, *Pfurtscheller and Lopes da Silva, 1999*; *Schroeder and Lakatos, 2009*). In the current study, our frequency-domain analysis makes no assumption about the increase and decrease of signals by face relative to non-face stimuli.

Finally, it is worth mentioning that previous studies have compared spatial properties across different signals using a slightly higher cutoff frequency for the high frequency range (>60 Hz in *Pfurtscheller et al., 2003*; >76Hz in *Miller et al., 2007*; >65Hz in *Hermes et al., 2012*) than the cutoff used here (>40 Hz). Using a higher frequency cutoff allows to limit contamination of high frequency

broadband signal by narrow-band gamma oscillations, which have different neurophysiological origins (*Ray and Maunsell, 2011*; *Hermes et al., 2015*), and are typically observed in the 30–80 Hz range. To ensure that our observations are not contaminated by putative narrow-band gamma oscillations (narrow-band gamma oscillations are thought to be attenuated when presenting natural images, *Hermes et al., 2015*), we replicated most analyses (including spatial extent) using the 80–160 Hz frequency range and obtained extremely similar observations compared to when using the 40–160 Hz (*Figure 5—figure supplement 4*; *Figure 5—figure supplement 5*).

In summary, by recording in a large area of the associative cortex, quantifying the rate of amplitude decay from the peak amplitude, and with no assumption regarding increase or decrease of signals, our study provides a more generalizable view of the relative local spatial extent of HF and LF signals than previous studies, pointing to comparable sizes of cortical sources generating the two types of signals.

## Corresponding functional properties across LF and HF face-selective activity

In addition to highly similar spatial properties of face-selectivity in VOTC for LF and HF signals, we provide three additional sources of evidence for a strong functional correspondence between these two signals. First, we found a robust correlation between face-selective amplitudes of LF and HF signals when using single recording contacts as unit data points ($r=0.63$ across the whole VOTC), with the highest correlation in the region showing the largest face-selective activity: the right latFG ($r=0.8$–$0.9$). As discussed above, only one human intracranial study (*Engell and McCarthy, 2011*) directly compared face-selective HF and LF (ERP) neural activities in the IOG and latFG, reporting a clearly lower amplitude correlation than here, but using different parameters to quantify face-selectivity across signals. We acknowledge that the correlations found here are not at ceiling and that there were also slight offsets in the location of maximum amplitude across signals along electrode arrays (*Figures 5 and 6*). This lack of a complete functional overlap between LF and HF is also in line with previous reports of slightly different selectivity and functional properties across these signals, such as a different sensitivity to spatial summation (*Winawer et al., 2013*), to selective attention (*Davidesco et al., 2013*) or to stimulus repetition (*Privman et al., 2011*). While part of these differences may be due to methodological differences in signal quantification, they also underline that these signals are not always strongly related, due to several factors. For instance, although both signals involve post-synaptic (i.e. dentritic) neural events, they nevertheless have distinct neurophysiological origins (that are not yet fully understood; see *Buzsáki et al., 2012*; *Leszczyński et al., 2020*; *Miller et al., 2009*) that are manifested in the current study as the difference in the frequency range at which these signals are most prevalent, or the difference of phase-locking relative to the stimulation between LF signals (here mostly phase-locked) and HF signals (mostly non phase-locked, *Lachaux et al., 2005*; *Winawer et al., 2013*). Future studies focusing on separating phase-locked and non-phase locked responses are needed to determine the respective role of frequency range vs. phase-locking in accounting for differences between these signals. In addition, these differing neurophysiological origins may interact with the precise setting of the recording sites capturing these signals (e.g. geometry/orientation of the neural sources relative to the recording site, cortical depth in which the signals are measured).

Second, using a face-selectivity index which takes into account the response to non-face objects (i.e. the general visual response), we found a similar magnitude of face-selectivity between LF and HF signals on overlapping contacts (LF+HF+). This observation is inconsistent with previous ECoG studies showing higher face-selectivity for HF than LF (measured as the N200) in the middle FG, but again using different quantification methods across signals (amplitude in a 40ms interval around N200 peak for ERP *vs.* power in the 0.1–0.35 s time-window for HF; *Rangarajan et al., 2014*) and underestimating face-selectivity of LF signals post-200 ms (e.g. *Engell and McCarthy, 2011*).

Third, focusing on the most responsive right latFG, we found similar timing onsets of face-selectivity between LF and HF signals (*Figure 8*). This similarity in onset latency is consistent with previous intra-cranial studies reporting the time-course of LF and HF responses to faces and non-face objects from the same electrodes in the IOG and latFG, although no quantitative and statistical comparisons were reported (*Lachaux et al., 2005*; *Engell and McCarthy, 2011*; *Rangarajan et al., 2014*). However, while none of these studies reported face-selectivity onset, our analyses filter-out the responses to non-face images (occurring at 6 Hz), thereby identifying onset latencies of *face-selective* activity for both signals at the same recording site for the first time.

An outstanding issue is how LF and HF activity respectively relate to behavior or perception. Both signals, when measured in VOTC, have been shown to be roughly equivalent in predicting the presentation time and category of a stimulus (face or house), although slightly higher performance were obtained when combining the signals, indicating that they contain a slight amount of complementary information (*Miller et al., 2016*). Other studies have also attempted to determine whether HF or LF correlate more with perceptual effects induced by local electrical stimulation of the cortex. For instance, *Winawer and Parvizi, 2016* found that HF receptive fields derived from visual retinotopic mapping are slightly more similar to size and locations of phosphenes induced by electrical stimulation, although a combination of signals was not tested. In addition, *Rangarajan et al., 2014* reported that conscious face distortions evoked by ECoG electrical stimulation over the middle FG could be predicted by the degree of face-selectivity measured in HF but not in LF, suggesting a higher functional value of HF than LF signals. Nevertheless, objective transient face identity recognition impairments during intracerebral electrical stimulation appear to be highly related to LF indexes of face-selectivity (*Jonas et al., 2012*; *Jonas et al., 2015*; see also *Jonas et al., 2018*; *Volfart et al., 2022* for similar relationships between LF and HF signals of face-selectivity and sensitivity to individual faces and the transient effect of intracerebral stimulation on behavior).

## Generalizability of our findings

An important issue is the degree to which our findings are generalizable to other brain functions and are comparable to previous findings obtained with different stimulation modalities or other recording methodologies. For several reasons, we argue that the approach used here (wide cortical sampling, SEEG recordings, neural responses to natural images of faces, frequency-tagging) provides good generalizability of our results. First, we recorded from a large cortical surface (i.e. the VOTC), ranging from the occipital to the temporal poles and encompassing various types of cortices, architectonics, and cognitive systems: primary and associative cortices, unimodal visual (posterior VOTC) and multi-modal semantic (ATL) cortices, neocortex and medial temporal structures (e.g. hippocampus). Given this diversity, our findings are likely to largely apply to other, unsampled brain regions. Second, we investigated category-selective neural responses to (human) faces, which are ubiquitous stimuli with high ecological validity in the human species known to recruit a widely distributed brain network (*Sergent et al., 1992*; *Haxby et al., 2000*; *Duchaine and Yovel, 2015*; *Gao et al., 2018*). Moreover, the use of natural images of such stimuli, as well as the contrasted nonface object categories, increase the ecological validity of our paradigm. Yet, whether our key findings of highly similar spatial distributions and functional properties of LF and HF signals extend to other recognition functions in vision or other sensory modalities remain to be tested in the future. Third, while ECoG mainly records activity from gyri and superficial cortical layers, SEEG (i.e. intracerebral) recordings, which are increasingly being used in the neuroscientific community, are sampled from all cortical layers and a wide range of anatomical structures and cortical geometries (sulci, gyri, deep structures, etc.), a diversity that increases generalizability of our findings. Despite this noticeable difference, the relatively low spatial resolution of these 2 recording methods (i.e., several millimeters) and size of recording sites (~2 mm) compared to the average cortical thickness (~2–3 mm) makes it very unlikely that SEEG and ECoG would reveal different patterns of LF-HF functional correspondence. Finally, we relied on a well-validated frequency-tagging stimulation approach, which provides objectivity of response identification and quantification, and similarity across analysis pipelines for LF and HF. Even though this approach remains relatively rare compared to slow temporally-jittered stimulation methods, streams of rapidly presented stimuli is a longstanding method of stimulation in human electrophysiology (*Adrian and Matthews, 1934*), widely used in vision (and also auditory and somatosensory stimulation), both in behavioral (e.g. rapid serial visual presentation- RSVP, e.g., *Potter and Levy, 1969*; *Potter et al., 2014*) and electrophysiological research ("steady-state potentials"; *Regan, 1966*). In addition, frequency-tagged neural responses as used in the present paradigm with natural variable images are independent of temporal predictability (i.e. strictly identical to responses obtained in a nonperiodic presentation mode; *Quek and Rossion, 2017*) and qualitatively similar across a well-defined suitable frequency range (*Retter and Rossion, 2016*; *Retter et al., 2020*), further increasing their generalizability.

## Conclusions

Overall, our large-scale intracerebral recording study shows that stimulus-triggered LF and HF signals recorded across the human associative cortex largely share spatial and functional properties, questioning a prevalent view in the neuroscientific community. Moreover, our observations indicate that neglecting LF signals in favor of HF activity comes at a cost in terms of identification and characterization of regions associated with low HF signal such as the VATL, which plays a key role in various higher order brain function including semantic cognition (*Lambon Ralph et al., 2010*; *Persichetti et al., 2021*). Although our observations were made with a specific type of stimulation in the visual modality and concerned essentially ventral occipital and temporal regions, the ubiquity of face stimuli in the natural environment, the wide cortical distribution dedicated to their recognition and their relevance for systems neuroscience in general, make it highly likely that these conclusions will generalize to other functional networks in the human brain.

## Materials and methods

### Participants

The study included 121 participants (61 females, mean age: 32.5±8.7 years; 109 right-handed) undergoing clinical intracerebral evaluation with depth electrodes (stereotactic electroencephalography or SEEG, *Figure 1A*) for refractory partial epilepsy. Participants were included in the study if they had at least one intracerebral electrode implanted in the ventral occipito-temporal cortex. The study includes the data from the 28 participants reported in *Jonas et al., 2016* and the 75 participants reported in *Hagen et al., 2020* for LF activity. All participants gave written consent to participate to the study, which was approved by the local human investigation committee (MAPCOG 2017-A03248-45).

### Fast periodic visual stimulation paradigm

A well validated FPVS paradigm with natural images was used to elicit high SNR face-selective neural activity in iEEG (see *Rossion et al., 2015* for the original description of the paradigm in EEG; *Rossion et al., 2018* for its validity in iEEG studies).

#### Stimuli

Two hundred grayscale natural images of various non-face objects (from 14 non-face categories: cats, dogs, horses, birds, flowers, fruits, vegetables, houseplants, phones, chairs, cameras, dishes, guitars, lamps) and 50 grayscale natural images of faces were used (the same stimuli as in *Rossion et al., 2015*; *Jonas et al., 2016*; *Hagen et al., 2020*). Each image contained an unsegmented object or face near the center, these stimuli differing in terms of size, viewpoint, lighting conditions and background. Global images were equalized for mean pixel luminance and contrast, but low-level visual cues associated with the faces and visual objects remained highly variable, naturally eliminating the systematic contribution of low-level visual cues to the recorded face-selective neural responses (*Rossion et al., 2015*; *Gao et al., 2018*).

#### Procedure

Participants viewed continuous sequences of natural images of objects presented at a fast rate of 6 Hz through sinusoidal contrast modulation. This relatively fast rate allows only one fixation per stimulus and is largely sufficient to elicit maximal face-selective activity (*Retter and Rossion, 2016*). Images of faces appear periodically as every 5th stimulus, so that neural activity that is common to faces and nonface stimuli is expressed at 6 Hz and harmonics, while differential (i.e. selective) responses to faces are expressed at 1.2 Hz (i.e. 6 Hz/5) (see *Figure 1B*). All images were randomly selected from their respective categories, with the constrain that no image could be immediately repeated. A stimulation sequence lasted 70 s: 66 s of stimulation at full-contrast flanked by 2 s of fade-in and fade-out, where contrast gradually increased or decreased, respectively. During a sequence, participants were instructed to fixate a small black cross which was presented continuously at the center of the stimuli and to detect brief (500ms) color-changes (black to red) of this fixation-cross. Among the 121 participants, participants viewed either 2 sequences (65 participants), 3 sequences (5 participants), 4 sequences (42 participants), 5 sequences (1 participant), 6 sequences (3 participants), 8, or more sequences (5 participants). These differences across participants were due to the particular

clinical context in which the experiment took place. Since the study concerned only group comparisons of the two types of intracranial EEG signals, all collected sequences were considered for analysis (i.e. the exact same sequences were compared across the two types of signals). No participant had seizures in the 2 hr preceding the recordings.

## Intracerebral electrode implantation and SEEG recording

Intracerebral electrodes (Dixi Medical, Besançon, France) were stereotactically implanted within the participants' brains for clinical purposes, that is, to delineate their seizure onset zones (*Talairach and Bancaud, 1973*) and to functionally map the surrounding cortex for potential epilepsy surgery (*Bédos Ulvin et al., 2017*). Each 0.8 mm diameter intracerebral electrode contains 5–15 independent recording contacts of 2 mm in length separated by 1.5 mm from edge to edge (*Figure 1A*, for details about the electrode implantation procedure, see *Salado et al., 2018*). The exact anatomical location of each recording contact was determined by coregistration of post-operative non-stereotactic CT-scan with a pre-operative T1-weighted MRI. A total of 856 electrode arrays were implanted in the VOTC of the 121 participants. These electrodes contained 9703 individual recording contacts in the VOTC (i.e., in the gray/white matter or medial temporal lobe-MTL; 5316 contacts in the left hemisphere, 4387 in the right hemisphere). Intracerebral EEG was sampled at either 500 or 512 Hz and referenced to either a midline prefrontal scalp electrode (FPz, 94 participants) or an intracerebral contact in the white matter (in 27 participants). SEEG signal at each recording contact was re-referenced offline to bipolar reference to limit dependencies between neighboring contacts (*Li et al., 2018*). Specifically, the signal at a given recording contact was computed as the signal measured at that contact (i.e. with the original recording reference) minus the signal at the directly adjacent contact located more medially on the same SEEG electrode array. Since SEEG field potentials are computed using pairs of adjacent contacts, each electrode array contains 1 contact less than in the original recording. All subsequent analyses were performed on bipolar-referenced signal in the set of bipolar contacts as described just above.

## SEEG signal processing and analyses

### Low-frequency (LF) analysis

SEEG signal processing for low frequency were largely similar to those in previous studies with this approach (*Jonas et al., 2016*; *Lochy et al., 2018*; *Hagen et al., 2020*). Segments of iEEG corresponding to stimulation sequences were extracted (74 s segments, –2 s to +72 s). The 74 s data segments were cropped to contain an integer number of 1.2 Hz cycles beginning 2 s after the onset of the sequence (right at the end of the fade-in period) until approximately 68 s, before stimulus fade-out (79 face cycles ≈ 65.8 s). Segments were averaged in the time-domain, then a Fast Fourier Transform (FFT) was applied to these averaged segments and amplitude spectra were extracted for all recording contacts (*Figure 1D and E*). The frequency-tagging approach used here allows identifying and separating two distinct types of responses (*Jonas et al., 2016*): (1) a general visual response occurring at the base stimulation frequency (6 Hz) and its harmonics, as well as (2) a face-selective activity at 1.2 Hz and its harmonics. Face-selective activity significantly above noise level at the face stimulation frequency (1.2 Hz) and its harmonics (2.4, 3.6 Hz, etc.) were determined as follows *Lochy et al., 2018*; *Jacques et al., 2020*: (1) the FFT spectrum was cut into 4 segments centered at the face frequency and harmonics, from the 1st until the 4th (1.2 Hz until 4.8 Hz), and surrounded by 25 neighboring bins on each side; (2) the amplitude values in these four segments of FFT spectra were summed; (3) the summed FFT spectrum was transformed into a Z-score. Z-scores were computed as the difference between the amplitude at the face frequency bin and the mean amplitude of 48 surrounding bins (25 bins on each side, excluding the 2 bins directly adjacent to the bin of interest, i.e. 48 bins) divided by the standard deviation of amplitudes in the corresponding 48 surrounding bins. A contact was considered as showing a face-selective response if the Z-score at the frequency bin of face stimulation exceeded 3.1 (i.e. $p < 0.001$ one-tailed: signal >noise).

### High-frequency (HF analysis)

Segments of iEEG corresponding to stimulation sequences were extracted (74-s segments, –2 s to +72 s). Variation in signal amplitude as a function of time and frequency was estimated by a Morlet wavelet transform applied on each SEEG segment from frequencies of 40–160 Hz, in 3 Hz increments

(*Figure 2D*, middle). Wavelet parameters were selected to ensure independence of the HF signal from higher frequency components of the LF evoked response (i.e. spectral leakage). The number of cycles (i.e. central frequency) of the wavelet was adapted as a function of frequency from 5 cycles at the lowest frequency to 11 cycles at the highest frequency, with a constant standard deviation in the time-domain of 0.15 s. The frequency bandwidth of the lowest wavelet (5 cycles at 40 Hz) was 20 Hz (full width at half maximum). Using these parameters, the lower boundary of the frequency bandwidth for the HF signal was 30 Hz (i.e. 40 – 20/2 Hz). This was well above the highest significant harmonic of face-selective response in the FPVS experiment which was 22.8 Hz (defined as the harmonic of the 1.2 Hz face frequency where, at the group level, the number of recording contacts with a significant response was not higher than the number of significant contacts detected for noise in bins surrounding harmonics of the face frequency). This ensures that the signal measured in the 40–160 Hz range is not contaminated by lower frequency responses. The wavelet transform was computed on each time-sample and the resulting amplitude envelope was downsampled by a factor of 6 (i.e. to 85.3 Hz sampling rate) to save disk space and computation time. For each segment, amplitude was normalized across time and frequency to obtain the percentage of signal change generated by the stimulus onset relative to the mean amplitude in a pre-stimulus time-window (–1600 ms to –300ms relative to the onset of the stimulation sequence). This normalization step ensures that each frequency in the HF range contributes equally to the computed HF signal, despite the overall $1/f$ relationship between amplitude and frequency in EEG. The percent signal change was then averaged across frequencies (between 30 Hz and 160 Hz) to obtain the time-varying HF amplitude envelope, and the segments were averaged in the time-domain (*Figure 2D*, bottom). The averaged segments were then cropped to contain an integer number of 1.2 Hz cycles (from 2 s after sequence onset to about 68.5 s, similarly as for LF) and the frequency content of the HF envelope was obtained using an FFT (*Figure 2F*). Significant face-selective responses in HF were detected based on the frequency spectra in exactly the same way as for the low-frequency bands. We used 30 Hz as the lower bound for HF, based on the observation that this frequency corresponds approximately to an intersection point on the frequency spectrum that separates properties of global power modulations above and below this frequency (*Miller et al., 2009*; *Podvalny et al., 2015*).

## Contact localization in the individual anatomy

The exact position of each contact relative to brain anatomy was determined in each participant's own brain by coregistration of the post-operative CT-scan with a T1-weighted MRI of the patient's head. Anatomical labels of bipolar contacts were determined using the anatomical location of the 'active' contact. In cases where the active contact was in the white matter and the 'reference' contact was in the gray matter, the active contact was labeled according to the anatomical location of the reference contact. Bipolar contacts in which both the active and reference contacts were in the white matter were excluded from amplitude and proportion analyses. To accurately assign an anatomical label to each contact, we used the same topographic parcellation of the VOTC as in *Jonas et al., 2016* and *Lochy et al., 2018* (*Figure 2—figure supplement 3*), which is close to the parcellation proposed by *Kim et al., 2000*.

## Classification of face-selective contacts according to LF and HF signals

For each VOTC contact, we determined whether there was a significant face-selective response in LF and in HF. This classification resulted in three non-overlapping sets of face-selective contacts: (1) LF+HF- (significant in LF but not in HF), (2) LF+HF+ (significant response in both signals), (3) LF-HF+ (significant only in HF). For analyses of proportions and amplitudes these contacts were sorted in two overlapping groups (see *Table 1*): LF+ (contacts exhibiting significant face-selective response in LF independently of HF) and HF+ (contacts exhibiting significant face-selective response in HF independently of LF).

## Quantification of response amplitudes

Amplitude quantification was performed on all face-selective contacts in the same manner for LF and HF responses. We first computed baseline-subtracted amplitudes in the frequency domain as the difference between the amplitude at each frequency bin and the average of 48 corresponding surrounding bins (up to 25 bins on each side, i.e. 50 bins, excluding the 2 bins directly adjacent to the

bin of interest, i.e. 48 bins). Then, for each contact, LF and HF face-selective amplitude was quantified as the sum of the baseline-subtracted amplitudes at the face frequency from the 1st until the 14th harmonic (1.2 Hz until 16.8 Hz), excluding the 5th and 10th harmonics (6 Hz and 12 Hz) that coincided with the base frequency (*Jonas et al., 2016*). Base response amplitude was quantified separately as the sum of the baseline-subtracted amplitudes at the base frequency from the 1st until the 3rd harmonic (6 Hz until 18 Hz). Contacts located in the same individually defined anatomical region (see section "Contact localization in the individual anatomy" below) were grouped, separately for the left and the right hemisphere. Since in all anatomical region, tails of the distribution of amplitudes contained values strongly outside of the normality range, amplitudes were winsorized (i.e. by clipping the largest and smallest 10% of amplitudes of the sample, *Dixon, 1960*; *Tukey, 1962*) by anatomical region before taking the mean and perform statistics. This allows providing an estimate of mean amplitude robust to these disproportionately large/small amplitudes. Then, for each anatomical region, we statistically compared group-level face-selective amplitude differences between hemispheres using general linear mixed effect models implemented in the lme4 package (*Bates et al., 2015*) in R v4.0.0. Statistical models were fitted using REML and p-values were obtained using Satterthwaite's approximation for degrees of freedom. Models included a random intercept per participant to account for both inter-participants variability and non-independence of contacts within a participant. P-values were corrected for multiple comparisons using Benjamini-Hochberg false discovery rate correction (*Benjamini and Hochberg, 1995*).

## Group visualization, proportion and amplitude analyses in Talairach space

For group analyses and visualization, anatomical MRIs were spatially normalized in order to determine the TAL coordinates of VOTC intracerebral contacts. The cortical surface used to display group contact locations and maps was obtained from segmenting the Collin27 brain from AFNI (*Cox, 1996*), which is aligned to the TAL space. We used TAL transformed coordinates to compute maps of the local proportion of face-selective intracerebral contacts across the VOTC. Proportions were computed separately for LF+ and HF+ contacts. Local proportion of contacts was computed in volumes (i.e. 'voxels') of size 12 x 12 x 100 mm (respectively for the X: left – right, Y: posterior – anterior, and Z: inferior – superior dimensions) by steps of 3 x 3 x 100 mm over the whole VOTC. A large voxel size in the Z dimension enabled collapsing across contacts along the inferior-superior dimension. For each voxel, we extracted the following information across all participants in our sample: (1) number of recorded contacts located within the voxel across all participants; (2) number of significant face-selective LF+ and HF+ contacts. From these values, for each voxel we computed the proportion of significant LF+ or HF+ contacts as the number of significant contacts within the voxel divided by the total number of recorded contacts in that voxel. Then, for each voxel we determined whether the proportions of significant LF+ or HF+ contacts were significantly above zero using a percentile bootstrap procedure, as follows: (1) within each voxel, sample as many contacts as the number of recorded contacts, with replacement; (2) for this bootstrap sample, determine the proportion of significant (LF+ or HF+) contacts and store this value; (3) repeat steps (1) and (2) 2000 times to generate a distribution of bootstrap proportions; and (4) estimate the p-value as the fraction of bootstrap proportions equal to zero. Following the same principle, we also visualized the variations in the proportions of LF+ and HF+ contacts as a function of the posterior-anterior axis (Y TAL dimension) collapsed across both hemispheres. Proportions were computed along the Y dimension in segments of 12 mm and by steps of 3 mm, collapsing contacts across the X (lateral-medial) and Z (inferior-superior) dimensions.

We also used the same mapping approach to compute the local LF or HF face-selective amplitude or face-selective Z-score across the VOTC. For amplitudes, we used the winsorized (see 'Quantification of response amplitudes') mean LF or HF baseline-subtracted amplitude across contacts within each 12 x 12 x 100 mm voxel over the cortical surface. Amplitude maps were computed either in LF+ or HF+ contacts, and Z-score maps were computed using all recorded VOTC contacts (i.e. whether contacts were face-selective or not).

## Correlation analyses

We performed a number of correlation analyses to estimate the similarity in the VOTC patterns of response amplitude/proportion for LF and HF signals and to quantify the functional correspondence between the two signals. Correlations were computed with either recording contacts or voxels (i.e.

mean over a set of contacts) as unit data points. We applied different methods to deal with the different shapes of distributions of the data. Since amplitude data was not normally distributed (skewed to the right), we applied a log transformation to normalize the data and applied a Pearson correlation on the log-transformed data. If the log transformation did not result in a normal distribution (e.g. when correlating maps of LF and HF proportions) we used the Spearman correlation which does not assume normality. In the result section, unless explicitly specified, correlations are performed using Pearson's r coefficient computed on non-normalized data.

We further characterized the magnitude of the correlations computed between LF and HF face-selective amplitude using individual recording contacts by comparing these correlations to the maximum correlation that is expected given the noise in the data (maximum expected correlation – MEC). The MEC, which derives from an estimation of test-retest reproducibility of the amplitudes, was computed by correlating the measured amplitudes to a simulated noisy measurement (modified from *Kay et al., 2013*). More specifically, first, we estimated the *noise* distribution by measuring the face-selective amplitudes for each stimulation sequence (in the exact same way as when using average across sequences as in the main analysis), taking the standard deviation of the amplitudes across stimulation sequences. This was done for each LF+HF+ contact and independently for LF and HF signals. Second, we used the amplitudes measured from averaged stimulation sequences (i.e. as in the main analyses) as an estimate of the 'true' *signal*. Third, using all LF+HF+ contacts, we determined the MEC using Monte-Carlo simulation, as follows. For each Monte-Carlo iteration (1) separately for LF and HF, we generate a simulated noisy set of amplitudes by summing the measured amplitudes (i.e. true signal) and simulated noise (computed by multiplying the noise distribution for each contact by random values drawn from a normal gaussian distribution); (2) we take the log of the measured and simulated amplitudes, and compute the Pearson correlation between these values; (3) we store the correlations (i.e. reflecting within-signal reproducibility) computed for LF and HF. We performed 2000 simulations, and averaged over the obtained 2000 correlation values for LF and HF to determine the mean reproducibility for each signal. The MEC was taken as the smaller of the two reproducibility values. This was done across all VOTC as well as separately for each main VOTC region.

## Estimating spatial extent of LF and HF signals

To estimate and compare the spatial extent of LF and HF signals at the millimeter scale we took advantage of the high spatial resolution of our SEEG setup in individual electrode arrays (i.e. 3.5 mm between the center location of recording contacts, see *Figure 1A*). We took into account face-selective responses in all bipolar contacts along these electrode arrays, irrespective of whether or not the contacts displayed a significant face-selective response and including contacts located in the white-matter. To ensure that the estimation of spatial extent was based on the same neural source for LF and HF signals in each electrode array, the analyses were restricted to electrodes that contain at least one LF+HF+ face-selective contact. For the same reason, we only included electrodes in which the contacts with the maximal face-selective amplitude for LF and HF were located within 3.5 mm of each other (which was 78% of the electrodes). We estimated the spatial extent of face-selective signals in the following manner, separately for LF and HF: (1) selected electrode arrays were grouped as a function of the hemisphere and the main VOTC region label (OCC, PTL, ATL) of the contact with the maximum amplitude (based on LF signal) in each electrode; (2) electrodes were spatially centered with respect to the contact with the largest amplitude; (3) for each electrode, amplitudes at contacts located on both sides of the maximum -and equidistant from it- were averaged in order to 'fold' the electrode around the maximum; (4) resulting response profile representing the variation in face-selective response amplitude as a function of the distance from the maximum amplitude in each electrode were averaged by region/hemisphere; (5) for each region/hemisphere, an exponential decay function ($y(x)=p * (1-d)^x+o$; x being the distance from peak; p, d and o being 3 parameters representing respectively the peak amplitude, decay constant and vertical amplitude offset) was fitted to the mean amplitude profile; (6) the spatial extent for LF and HF signals in each region was quantified as the distance from the maximum at which the exponential decay function reached half of its amplitude range. Permutation tests (2000 permutations) were used to statistically compare the spatial extent across LF and HF in each region/hemisphere.

## Face-selectivity index

We computed an index of face-selectivity for LF and HF signals by taking the ratio of the face-selective amplitude (i.e. at 1.2 Hz and harmonics) to the sum of the face-selective and general visual (i.e. amplitude at 6, 12, 18 Hz) responses. This index provides an additional quantification of face-selectivity by taking into account the magnitude of response to non-face stimuli. Since the index varies from 0 (no face-selective response) to 1 (face-selective response only, no general visual response) it also allows to directly compare face-selectivity across LF and HF. We computed the face-selectivity index either at the scale of main VOTC regions or by mapping the index using contacts localized in TAL space. In both cases, the face-selective and general visual responses were first averaged across contacts (i.e. within regions or voxels) before computing the index. This avoided obtaining indices outside the [0–1] range if, for instance, the denominator is smaller than 1.

## Timing of face-selectivity

For recording contacts located in the right latFG where responses were maximal for both signals, we also compared LF and HF activity in the time-domain. To increase the temporal resolution of the HF signal for the timing analysis, we recomputed the HF signal using a number of cycles for the wavelet that ranged from 3.2 cycles (at 40 Hz) to 9 cycles (at 160 Hz). This resulted in wavelet duration of 20–30ms (i.e. full width at half maximum). The lower boundary of the frequency bandwidth for the HF signal using these parameters was 24.4 Hz which was still above the highest significant harmonic of face-selective response in the FPVS experiment (22.8 Hz). We only used LF+HF+ contacts (N=66 in right latFG) since they exhibit significant responses for both signals. To account for the sinusoidal modulation of contrast (rather than an abrupt onset; *Figure 1A*), the face onset time was shifted forward by 41ms (~1/4 of a 6 Hz cycle duration), corresponding to a face contrast of about 50% of the maximal (*Retter and Rossion, 2016*). The starting point for the analyses in the time-domain was, for LF: the raw 70 s time-series of EEG recorded for each FPVS sequence; and for HF: the 70 s HF (i.e. average over 40–160 Hz) amplitude modulation time-series. Then, for each recording contact, LF and HF time-series were processed in the following way: (1) an FFT notch filter (filter width = 0.07 Hz) was applied to remove the general visual response at 6 Hz and three additional harmonics (i.e. 6, 12, 18, 24 Hz) and, specifically for the LF signal, an additional low-pass filter to remove signal above 30 Hz (Butterworth filter order 4); (2) time-series were segmented in 1.17 s epochs centered on the onset of each face (i.e. [–2 to +5] 6Hz-cycles relative to face onset) in the FPVS sequences; (3) resulting epochs were averaged and (4) baseline-corrected by subtracting the mean amplitude in a [-0.166–0 s] time-window relative to face onset. The resulting averaged time-domain responses per recording contact were used to determine the response latency of the face-selective response at each contact. Response onset latency was computed as the time-point at which the EEG response was outside the interval defined by the mean amplitude measured in an interval before face onset (i.e. [-0.166–0 s]) +/-2.58 times the standard deviation of the amplitude in the same interval (i.e. p<0.01, two-tailed), for at least 30ms. We used these latencies to compute the Pearson correlation between the onset latencies of face-selective responses for LF *vs.* HF. Additionally, the latency of LF and HF face-selective response onset and offset at the group level were determined on the responses averaged across recording contacts in the latFG. For LF, to limit the variability in the response morphology or polarity across recording contacts, the averaged response at each contact was first transformed into phase-free variations of amplitude over time using a Hilbert transform, before averaging across contact. Onset and offset latencies for each signal was estimated using a percentile bootstrap approach in the following way: (1) randomly sample contacts with replacement, (2) average (mean) responses from sampled contacts, (3) repeat steps (1) and (2) 1000 times, (4) based on step (3), compute a p-value for each time point as the fraction of bootstrap samples higher than the mean amplitude in the pre-stimulus baseline, (5) determine response latency as the time-point at which p<0.01 for at least 30 consecutive ms and store this latency value, (6) repeat steps (1) to (5) 1000 times to obtain a distribution and confidence interval for the response latencies.

## Acknowledgements

We thank the participants for their involvement in the study. This work was supported by an EOS grant "HUMVISCAT" 30991544.

# Additional information

## Funding

| Funder | Grant reference number | Author |
|---|---|---|
| Belgian Excellence of Science program | "HUMVISCAT" 30991544 | Bruno Rossion |

The funders had no role in study design, data collection and interpretation, or the decision to submit the work for publication.

## Author contributions

Corentin Jacques, Conceptualization, Software, Formal analysis, Investigation, Visualization, Methodology, Writing – original draft, Writing – review and editing; Jacques Jonas, Conceptualization, Resources, Formal analysis, Investigation, Visualization, Methodology, Writing – original draft, Writing – review and editing; Sophie Colnat-Coulbois, Resources, Investigation, Project administration; Louis Maillard, Resources, Funding acquisition, Project administration; Bruno Rossion, Conceptualization, Resources, Supervision, Funding acquisition, Investigation, Methodology, Writing – original draft, Project administration, Writing – review and editing

## Author ORCIDs

Corentin Jacques http://orcid.org/0000-0001-8917-4346
Jacques Jonas http://orcid.org/0000-0002-1053-0748
Bruno Rossion http://orcid.org/0000-0002-1845-3935

## Ethics

Human subjects: All participants gave written consent to participate to the study, as approved by the human investigation committee (CHRU Nancy, reference center; MAPCOG-SEEG 2017-A03248-45).

## Decision letter and Author response

Decision letter https://doi.org/10.7554/eLife.76544.sa1
Author response https://doi.org/10.7554/eLife.76544.sa2

# Additional files

## Supplementary files
• Transparent reporting form

## Data availability

All SEEG data generated or analysed during this study have been uploaded to Dyad (https://doi.org/10.5061/dryad.66t1g1k51).

The following dataset was generated:

| Author(s) | Year | Dataset title | Dataset URL | Database and Identifier |
|---|---|---|---|---|
| Rossion B | 2022 | Data from: Low and high frequency intracranial neural signals match in the human associative cortex | http://dx.doi.org/10.5061/dryad.66t1g1k51 | Dryad Digital Repository, 10.5061/dryad.66t1g1k51 |

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
