## [Editor Report]

This is an important paper that will be of great interest to researchers interested in neural brain signals at different frequencies. It shows that low-frequency local field potentials and high-frequency (>30 Hz) broadband activity in response to face stimuli have largely similar spatial, functional, and timing properties. The compelling findings are supported by an innovative paradigm and analysis of intracranial recordings in 121 human participants. These observations provide novel basic science insights into how brain responses at different frequencies signal sensory information.

---

## [Decision Letter]

**Decision letter after peer review:**

Thank you for submitting your article "Low and high frequency intracranial neural signals match in the human associative cortex" for consideration by *eLife*. Your article has been reviewed by 3 peer reviewers, including Markus Ploner as Reviewing Editor and Reviewer #1, and the evaluation has been overseen by Christian Büchel as the Senior Editor.

Essential revisions:

1) It is essential that the authors demonstrate that the low and high-frequency signals do not confound each other but represent different types of signals. In particular, it should be ruled out that the high-frequency signals are confounded by spectral leakage of the response amplitude envelope, the type of time-frequency calculation, the noise level, or 1/f contributions. Subtracting the evoked signal from the broadband signal and excluding the lower frequency end of the broadband range might be important control analyses to address these issues.

2) It remains unclear whether and how the current findings generalize to the processing of other sensory stimuli and paradigms. Rhythmic presentation of visual stimuli at 6 Hz with face stimuli every five stimuli (1.2 Hz) represents a very particular type of sensory stimulation. Stimulation with other stimuli, or at other frequencies likely induce different responses. Moreover, signals from depth electrodes might differ from those recorded by surface electrodes. These important limitations should be appropriately discussed.

*Reviewer #1 (Recommendations for the authors):*

– The authors should use a more neutral and balanced writing style that avoids emphasizing a "race" between the functional significance of LF and HF brain responses and researchers focusing on one or the other.

– The authors should acknowledge that it remains unclear whether and how the current findings generalize to the processing of other sensory stimuli and paradigms. This important limitation should be explicitly mentioned in the abstract and appropriately discussed in the Discussion section.

– P.12, line 1: It should read "Figure 4A" instead of "Figure 3A".

– P.14, line 1: Please clarify "Figure SX".

*Reviewer #2 (Recommendations for the authors):*

1) The authors frequently refer to the signals as differing in frequency (low vs high), which is only one way in which the signals differ: they also differ in that the low-frequency signal is phase-locked and the high-frequency signal is (probably) not. More concretely, the low-frequency signal is first order (the amplitude of voltage fluctuations), and the high-frequency signal is second order (the amplitude of fluctuations). This other difference is clear in the introduction but then mostly dropped for the rest of the paper. Any differences found (such as in SNR) could, in principle, either be due to the difference in frequency range or in first vs second order. For example, if stimuli had been shown at a higher frequency, there would be an evoked signal at higher frequencies (esp considering the harmonics). Additionally, there might be non-phase-locked signals at lower frequencies in this or other experiments. Whether the higher SNR would depend on frequency or phase-locking or both is not addressed by this study. This is not a problem per se, but as the paper is written now, the frequent references to low vs high frequency implicitly embrace only one of two possible interpretations. A brief acknowledgement of these possibilities, either in Discussion or Results would clarify the issue to the reader.

2) Some studies of broadband signals subtract (or project out) the evoked signal in order to isolate the induced (non-phase locked) signal from the evoked (phase-locked) signal. Doing so, at least as a control analysis, would strengthen the conclusion. As it is presented now, the possible confound is that the two signals look similar because they are both largely reflecting the evoked signal. While this is unlikely to be the explanation for the similarity of the two signals, it is easy enough to check and rule out, and given the relatively bold conclusions, ruling out possible confounds is important.

3) Related to 2, the cut-off of 30 Hz at the lower end of the broadband signal makes it possible that the broadband measure includes γ oscillations, which may differ from the high-frequency broadband signals. Many of the claims about the spatial specificity of high-frequency signals are based on analyses of responses at a higher frequency than γ oscillations (eg 80 Hz and above, or 100 Hz and above). This is not because the broadband signal is only found at the higher frequency range, but because the lower end (say, 30 Hz to 80 Hz) might include both. The results would be more convincing if they were shown to be similar when excluding the lower frequency end of the broadband range. (Note that this would also decrease the change of "contamination" of the broadband signal by the evoked signal).

*Reviewer #3 (Recommendations for the authors):*

– Regarding the high number of participants: the given citation is from 2007, in the last years there have been quite a number of intracranial studies with a large number of participants (e.g. some random ones: Quon et al. 2021 n=261, Frauscher et al. 2018 n=106, Mohan et al. 2022 n=145 biorxiv), while the number of participants in the present study is definitely high, the extent of it is overstated in the current text.

---

## [Author Response]

Essential revisions:1) It is essential that the authors demonstrate that the low and high-frequency signals do not confound each other but represent different types of signals. In particular, it should be ruled out that the high-frequency signals are confounded by spectral leakage of the response amplitude envelope, the type of time-frequency calculation, the noise level, or 1/f contributions. Subtracting the evoked signal from the broadband signal and excluding the lower frequency end of the broadband range might be important control analyses to address these issues.

In the revised manuscript we have taken several steps to ensure the full independence of low and high-frequency signals.

First, we recomputed the HF signal using time-frequency analyses (wavelet) parameters with a smaller frequency bandwidth (Half width at half maxima of the wavelets used in the revised analysis varies between 10 and 15Hz) and changed the frequency range to 40-160 Hz (instead of 30-160 Hz initially). This ensures that the lowest frequency included in the HF signal was 30 Hz. This is well above the highest significant harmonic of the low-frequency face-selective response in our frequency-tagging experiment, which was 22.8 Hz. Thus, the signal measured in the 40-160 Hz range is not contaminated by lower frequency evoked responses. We recomputed all analyses and statistics from the manuscript with the new HF definition. Overall, this change had very little impact on the findings. This indicates that the HF activity was mostly independent from phase-locked LF signal already in the original submission and that our initial conclusions were correct. However, since the analyses with the revised time-frequency analyses parameters enforces this independence, the revised analyses have become the main analyses in the manuscript. The manuscript has been revised accordingly and all figures (main and supplementary) have been modified to reflect the new analyses.

Second, as suggested by reviewer 2, we conducted supplemental analyses by restricting the HF signal to the 80-160 Hz range (please see below).

The 1/f decrease of amplitude as a function of frequency, which gives more weight to low frequencies when averaging signals within a given frequency range, was dealt with in the original (and revised) submission by computing the percent signal change relative to the pre-stimulation baseline independently for each frequency bin, BEFORE averaging across the frequencies in the HF range (i.e., 40-160Hz). This ensures that each frequency contributes equally to the computed HF signal, despite the overall 1/f shape of the power spectrum. The following sentence was added in the method section (‘HF analysis’), p 38 (line 1038):

“This normalization step ensures that each frequency in the HF range contributes equally to the computed HF signal, despite the overall 1/f relationship between amplitude and frequency in EEG.”

In addition, we did not subtract the evoked response prior to computing the HF signal since this approach can produce artifactual responses. Indeed, there are known trial-to-trial variations in the amplitude and latency of the LF phase-locked responses. Therefore, subtracting the averaged evoked response from each trial may generate spurious and unpredictable signals, particularly at higher frequency in case of small latency variations, which could themselves produce artificial contamination of HF signal by LF evoked activity (see e.g., Truccolo et al., 2002; see list of references at the end of the reply).

2) It remains unclear whether and how the current findings generalize to the processing of other sensory stimuli and paradigms. Rhythmic presentation of visual stimuli at 6 Hz with face stimuli every five stimuli (1.2 Hz) represents a very particular type of sensory stimulation. Stimulation with other stimuli, or at other frequencies likely induce different responses. Moreover, signals from depth electrodes might differ from those recorded by surface electrodes. These important limitations should be appropriately discussed.

We agree that the issue of the generalizability of our results to other brain functions and to previous findings obtained with different stimulation modalities or other recording methodologies is important. We explicitly discuss this issue in the revised version of the manuscript, in particular by adding a new section in the Discussion section (p. 32-33). We argue that our original methodological approach allows maximizing the generalizability of our findings. First, we recorded form a large cortical surface. Second, we investigated responses to ubiquitous stimuli for the human brain, i.e., faces, which recruit a wide network in visual and memory systems. Third, we used depth electrodes, sampling all cortical layers and a wide range of cortical and non-cortical structures. Finally, we recorded responses through a frequency-tagging approach that increases the reliability of our observations (thanks to objective response identification and quantification and similar analysis pipelines across signal types). Based on previous research, we also indicate that this periodic mode of visual presentation does not lead to atypical brain responses (i.e., in fact, the responses are identical to nonperiodic visual presentation; Quek and Rossion, 2017) and that the responses are independent from the frequency of presentation, as long as the base rate frequency is within the suitable range for the studied function (Retter et al., 2021).

Here is the section added to the discussion:

**“**Generalizability of our findings

An important issue is the degree to which our findings are generalizable to other brain functions and are comparable to previous findings obtained with different stimulation modalities or other recording methodologies. […] In addition, frequency-tagged neural responses as used in the present paradigm with natural variable images are independent of temporal predictability (i.e., strictly identical to responses obtained in a nonperiodic presentation mode; Queck and Rossion, 2017) and qualitatively similar across a well-defined suitable frequency range (Retter and Rossion, 2016; Retter et al., 2020), further increasing their generalizability.”

Reviewer #1 (Recommendations for the authors):– The authors should use a more neutral and balanced writing style that avoids emphasizing a "race" between the functional significance of LF and HF brain responses and researchers focusing on one or the other.– The authors should acknowledge that it remains unclear whether and how the current findings generalize to the processing of other sensory stimuli and paradigms. This important limitation should be explicitly mentioned in the abstract and appropriately discussed in the Discussion section.

Please see our reply above about the addition of a section in the revised discussion.

– P.12, line 1: It should read "Figure 4A" instead of "Figure 3A".– P.14, line 1: Please clarify "Figure SX".

These were corrected, thank you.

Reviewer #2 (Recommendations for the authors):1) The authors frequently refer to the signals as differing in frequency (low vs high), which is only one way in which the signals differ: they also differ in that the low-frequency signal is phase-locked and the high-frequency signal is (probably) not. More concretely, the low-frequency signal is first order (the amplitude of voltage fluctuations), and the high-frequency signal is second order (the amplitude of fluctuations). This other difference is clear in the introduction but then mostly dropped for the rest of the paper. Any differences found (such as in SNR) could, in principle, either be due to the difference in frequency range or in first vs second order. For example, if stimuli had been shown at a higher frequency, there would be an evoked signal at higher frequencies (esp considering the harmonics). Additionally, there might be non-phase-locked signals at lower frequencies in this or other experiments. Whether the higher SNR would depend on frequency or phase-locking or both is not addressed by this study. This is not a problem per se, but as the paper is written now, the frequent references to low vs high frequency implicitly embrace only one of two possible interpretations. A brief acknowledgement of these possibilities, either in Discussion or Results would clarify the issue to the reader.

We agree with the reviewer’s point. Therefore, in the introduction we have more clearly defined the signals of interest, and added a detailed acknowledgement of the relationship between frequency range and phase-locking as a footnote in the introduction.

Here are the modified sentences in the introduction:

“At a macroscopic level of organization, there are two prominent neurophysiological signals. On the one hand, event-related potentials (ERPs; often called local field potentials in iEEG) which are time-locked and largely phase-locked to an event (e.g., a sensory stimulus in the visual modality) and predominant in the lower range of the frequency spectrum (< 30Hz, here referred to as ‘low frequency’ activity, LF). On the other hand, broadband activity which is largely non phase-locked relative to events and typically observed and quantified over a higher frequency range of the spectrum (>30Hz also known as ‘γ range’; here ‘high frequency’ activity, HF).”

As well as the added footnote in the introduction:

“LF and HF signals differ both in the frequency range at which they are prevalent, and in terms of their phase-locking relative to the event onset. […] Future studies focusing on separating phase-locked and non-phase locked responses are needed to determine the respective role of frequency range vs. phase-locking in accounting for differences between these signals.”

2) Some studies of broadband signals subtract (or project out) the evoked signal in order to isolate the induced (non-phase locked) signal from the evoked (phase-locked) signal. Doing so, at least as a control analysis, would strengthen the conclusion. As it is presented now, the possible confound is that the two signals look similar because they are both largely reflecting the evoked signal. While this is unlikely to be the explanation for the similarity of the two signals, it is easy enough to check and rule out, and given the relatively bold conclusions, ruling out possible confounds is important.

First, given that HF signal is largely broadband and measured in a high-frequency window (40-160 Hz; e.g., Figure 1; i.e. above the frequency at which highest significant LF face-selective evoked responses are observed in our study, ~ 23 Hz), there is little reason to think that the HF activity would reflect an evoked (i.e. phase-locked) signal, and no reason to consider that it could be more so than in other iEEG studies that have investigated this signal and/or compared it to LF signals (e.g. Engell and McCarthy, 2011). Thus, we consider that our comparison of the two signals is in line with what has been done in previous studies (with the improvements that we introduce in the paper in terms of similarity of analysis and objectivity of identification and quantification of signal).

Second, we are familiar with the method mentioned by the reviewed which consists in averaging single trials to derive the mean evoked response and then subtract or regress-out this average response from each single trial (Kalcher and Pfurtsheller, 1995; Cohen and Donner, 2013). However, this approach is problematic for two reasons:

First, this approach can produce artifactual responses. Indeed, there are known small trial-to-trial variations in the amplitude and latency of the LF phase-locked responses. Therefore, subtracting the averaged evoked response from each trial may generate spurious and unpredictable signals, particularly at higher frequency in case of small latency variations, which could themselves produce artificial contamination of HF signal by LF evoked activity (see e.g., Truccolo et al., 2002).

Second, the subtraction procedure is not really compatible with our quantification in the frequency –domain. Indeed, to apply the procedure, we would have to segment the long (~60 seconds) recordings sequences in segments of 1 face-cycle duration ( = 1/1.2 = 0.833 s), average those segments in the time-domain to generate an average evoked response, subtract this average 0.833s segment from the original sequences at each face onset (alternatively one could re-construct artificial long sequences by stitching together the average evoked response of 0.833s and subtract it from the original sequences). The simple fact of subtracting successively the same signal of 0.833 s duration (i.e. 1.2Hz) from the original sequences might generate an unpredictable signal at 1.2Hz and harmonics (due to potential edges created, due to the subtraction, every 0.833 s in the resulting signal).

For these reasons we opted not to perform this potentially problematic analysis.

3) Related to 2, the cut-off of 30 Hz at the lower end of the broadband signal makes it possible that the broadband measure includes γ oscillations, which may differ from the high-frequency broadband signals. Many of the claims about the spatial specificity of high-frequency signals are based on analyses of responses at a higher frequency than γ oscillations (eg 80 Hz and above, or 100 Hz and above). This is not because the broadband signal is only found at the higher frequency range, but because the lower end (say, 30 Hz to 80 Hz) might include both. The results would be more convincing if they were shown to be similar when excluding the lower frequency end of the broadband range. (Note that this would also decrease the change of "contamination" of the broadband signal by the evoked signal).

This is an interesting point. In the literature, there is no consensus about what should be the lower bound of the high frequency broadband or γ range. It ranges from 30 Hz to about 80 Hz. We originally used 30 Hz as a lower bound for HF, based on the observation that this frequency corresponds approximately to an intersection point on the frequency spectrum that separates properties of global power modulations above and below this frequency (Miller et al., 2009; Podvalny et al., 2015). Other researchers divide the broadband/γ range into high and low γ range around 80 Hz (e.g. Ray and Maunsell, 2011; Foster an Parvizi, 2012).

It is unlikely that the signal in the current experiment contains strong γ ‘oscillations’, since these oscillations are mostly absent when using complex stimuli such as those used here (Hermes et al., Cerebral Cortex, 2014). Nevertheless, to take into account for the reviewer’s comment, we re-computed most of the analyses using the 80-160 Hz frequency range. We also computed correlations between HF 40-160 Hz and HF 80-160 Hz. This is included as supplementary material (Figure 5 —figure supplement 4; Figure 5 —figure supplement 5). We found a very high correlation between the signals in the two frequency ranges (r = 0.98 over HF+ contacts). Moreover, the overall pattern of results is replicated even with this much narrower frequency range, including the similarity in the spatial extent of the LF and HF signals.

This was added to the Results, in the section about spatial extent (p. 19, line 465) : “We obtained almost identical observations when restricting HF signal to the high-γ range, i.e. 80-160Hz (Figure 5, figure supplement 4; Figure 5, figure supplement 5).” In addition, we briefly discuss this issue in the discussion, in the section on spatial extent (p. 30):

“Finally, it is worth mentioning that previous studies that have compared spatial properties across different signals using a slightly higher cutoff frequency for the high frequency range (>60Hz in Pfurtscheller et al., 2003; >76Hz in Miller et al., 2007; >65Hz in Hermes et al., 2012) than the cutoff used here (>40Hz). Using a higher frequency cutoff allows to limit contamination of high frequency broadband signal by narrow-band γ oscillations, which have different neurophysiological origin (Ray and Maunsell, 2011; Hermes et al., 2014), and are typically observed in the 30-80 Hz range. To ensure that our observations are not contaminated by putative narrow-band γ oscillations (narrow-band γ oscillation are thought to be attenuated when presenting natural images, Hermes et al., 2014), we replicated most analyses (including spatial extent) using the 80 to 160 Hz frequency range and obtained extremely similar observations compared to when using the 40 to 160 Hz (Figure 5—figure supplement 4; Figure 5—figure supplement 5).”

Reviewer #3 (Recommendations for the authors):– Regarding the high number of participants: the given citation is from 2007, in the last years there have been quite a number of intracranial studies with a large number of participants (e.g. some random ones: Quon et al. 2021 n=261, Frauscher et al. 2018 n=106, Mohan et al. 2022 n=145 biorxiv), while the number of participants in the present study is definitely high, the extent of it is overstated in the current text.

We focused specifically on intracranial EEG studies recording the evoked electrophysiological responses to a stimulus presentation or an active cognitive task. Among the articles cited by the reviewer, only one corresponds to this type of studies (Mohan et al., 2022). We honestly believe that it is not an exaggeration to write that these kind of intracranial EEG studies with more than 100 participants are very rare currently (even if such studies included more and more participants in the recent years, but usually below 100).

References:

Adrian, E., Matthews, B., 1934. The Berger rhythm: potential changes from the occipital lobes in man. Brain 57, 355–385.

Alonso-Prieto, E., Van Belle, G., Liu-Shuang, J., Norcia, A.M., Rossion, B., 2013. The 6 Hz fundamental stimulation frequency rate for individual face discrimination in the right occipito-temporal cortex. Neuropsychologia 51, 2863–2875. https://doi.org/10.1016/j.neuropsychologia.2013.08.018

Bastin, J., Vidal, J.R., Bouvier, S., Perrone-Bertolotti, M., Bénis, D., Kahane, P., David, O., Lachaux, J.-P., Epstein, R. a, 2013. Temporal components in the parahippocampal place area revealed by human intracerebral recordings. J. Neurosci. 33, 10123–10131. https://doi.org/10.1523/JNEUROSCI.4646-12.2013

Cohen, M.X., Donner, T.H., 2013. Midfrontal conflict-related theta-band power reflects neural oscillations that predict behavior. J. Neurophysiol. 110, 2752–2763. https://doi.org/10.1152/jn.00479.2013

Crone, N.E., Boatman, D., Gordon, B., Hao, L., 2001. Induced electrocorticographic γ activity during auditory perception. Brazier Award-winning article, 2001. Clin. Neurophysiol. Off. J. Int. Fed. Clin. Neurophysiol. 112, 565–582. https://doi.org/10.1016/s1388-2457(00)00545-9

Crone, N.E., Miglioretti, D.L., Gordon, B., Lesser, R.P., 1998. Functional mapping of human sensorimotor cortex with electrocorticographic spectral analysis. II. Event-related synchronization in the γ band. Brain 121 Pt 1, 2301–2315. https://doi.org/10.1093/brain/121.12.2301

Davidesco, I., Harel, M., Ramot, M., Kramer, U., Kipervasser, S., Andelman, F., Neufeld, M.Y., Goelman, G., Fried, I., Malach, R., 2013. Spatial and object-based attention modulates broadband high-frequency responses across the human visual cortical hierarchy. J. Neurosci. 33, 1228–1240. https://doi.org/10.1523/JNEUROSCI.3181-12.2013

Flinker, A., Chang, E.F., Barbaro, N.M., Berger, M.S., Knight, R.T., 2011. Sub-centimeter language organization in the human temporal lobe. Brain Lang. 117, 103–109. https://doi.org/10.1016/j.bandl.2010.09.009

Flinker, A., Korzeniewska, A., Shestyuk, A.Y., Franaszczuk, P.J., Dronkers, N.F., Knight, R.T., Crone, N.E., 2015. Redefining the role of Broca’s area in speech. Proc. Natl. Acad. Sci. 112, 2871–2875. https://doi.org/10.1073/pnas.1414491112

Foster, B.L., Parvizi, J., 2012. Resting oscillations and cross-frequency coupling in the human posteromedial cortex. Neuroimage 60, 384–391. https://doi.org/10.1016/j.neuroimage.2011.12.019

Golan, T., Davidesco, I., Meshulam, M., Groppe, D.M., Mégevand, P., Yeagle, E.M., Goldfinger, M.S., Harel, M., Melloni, L., Schroeder, C.E., Deouell, L.Y., Mehta, A.D., Malach, R., 2017. Increasing suppression of saccade-related transients along the human visual hierarchy. e*Life* 6, e27819. https://doi.org/10.7554/*eLife*.27819

Golan, T., Davidesco, I., Meshulam, M., Groppe, D.M., Mégevand, P., Yeagle, E.M., Goldfinger, M.S., Harel, M., Melloni, L., Schroeder, C.E., Deouell, L.Y., Mehta, A.D., Malach, R., 2016. Human intracranial recordings link suppressed transients rather than ’filling-in’ to perceptual continuity across blinks. e*Life* 5, 1–28. https://doi.org/10.7554/*eLife*.17243

Hagen, S., Jacques, C., Maillard, L., Colnat-Coulbois, S., Rossion, B., Jonas, J., 2020. Spatially Dissociated Intracerebral Maps for Face- and House-Selective Activity in the Human Ventral Occipito-Temporal Cortex. Cereb. Cortex 30, 4026–4043. https://doi.org/10.1093/cercor/bhaa022

Hermes, D., Miller, K.J., Wandell, B.A., Winawer, J., 2015. Stimulus Dependence of Γ Oscillations in Human Visual Cortex. Cereb. cortex 25, 2951–2959. https://doi.org/10.1093/cercor/bhu091

Jacques, C., Retter, T.L., Rossion, B., 2016. A single glance at natural face images generate larger and qualitatively different category-selective spatio-temporal signatures than other ecologically-relevant categories in the human brain. Neuroimage 137, 21–33. https://doi.org/10.1016/j.neuroimage.2016.04.045

Jonas, J., Jacques, C., Liu-shuang, J., Brissart, H., Colnat-coulbois, S., Maillard, L., Rossion, B., 2016. A face-selective ventral occipito-temporal map of the human brain with intracerebral potentials. Proc. Natl. Acad. Sci. U. S. A. 113, E4088--E4097. https://doi.org/10.1073/pnas.1522033113

Kadipasaoglu, C.M., Conner, C.R., Baboyan, V.G., Rollo, M., Pieters, T.A., Tandon, N., 2017. Network dynamics of human face perception. PLoS One 12, e0188834. https://doi.org/10.1371/journal.pone.0188834

Kadipasaoglu, C.M., Conner, C.R., Whaley, M.L., Baboyan, V.G., Tandon, N., 2016. Category-Selectivity in Human Visual Cortex Follows Cortical Topology : A Grouped icEEG Study 1–28. https://doi.org/10.1371/journal.pone.0157109

Kalcher, J., Pfurtscheller, G., 1995. Discrimination between phase-locked and non-phase-locked event-related EEG activity. Electroencephalogr. Clin. Neurophysiol. 94, 381–384. https://doi.org/https://doi.org/10.1016/0013-4694(95)00040-6

Kamp, A., Sem Jacobsen, C.W., Storm Van Leeuwen, W., van der Tweel, L., 1960. Cortical responses to modulated light in the human subject. Acta Physiol. Scand. 48, 1–12. https://doi.org/10.1111/j.1748-1716.1960.tb01840.x

Manning, J.R., Jacobs, J., Fried, I., Kahana, M.J., 2009. Broadband shifts in local field potential power spectra are correlated with single-neuron spiking in humans. J. Neurosci. 29, 13613–13620. https://doi.org/10.1523/JNEUROSCI.2041-09.2009

Mesgarani, N., Chang, E.F., 2012. Selective cortical representation of attended speaker in multi-talker speech perception. Nature 485, 233–236. https://doi.org/10.1038/nature11020

Miller, K.J., Sorensen, L.B., Ojemann, J.G., den Nijs, M., 2009. Power-law scaling in the brain surface electric potential. PLoS Comput. Biol. 5, e1000609. https://doi.org/10.1371/journal.pcbi.1000609

Nir, Y., Fisch, L., Mukamel, R., Gelbard-Sagiv, H., Arieli, A., Fried, I., Malach, R., 2007. Coupling between Neuronal Firing Rate, Γ LFP, and BOLD fMRI Is Related to Interneuronal Correlations. Curr. Biol. 17, 1275–1285. https://doi.org/10.1016/j.cub.2007.06.066

Norcia, A.M., Appelbaum, L.G., Ales, J.M., Cottereau, B.R., Rossion, B., 2015. The steady-state visual evoked potential in vision research: A review. J. Vis. 15, (6) 4, 1--46. https://doi.org/10.1167/15.6.4

Pfurtscheller, G., Graimann, B., Huggins, J.E., Levine, S.P., Schuh, L.A., 2003. Spatiotemporal patterns of β desynchronization and γ synchronization in corticographic data during self-paced movement. Clin. Neurophysiol. 114, 1226–1236. https://doi.org/10.1016/S1388-2457(03)00067-1

Podvalny, E., Noy, N., Harel, M., Bickel, S., Chechik, G., Schroeder, C.E., Mehta, A.D., Tsodyks, M., Malach, R., 2015. A unifying principle underlying the extracellular field potential spectral responses in the human cortex. J. Neurophysiol. 114, 505–519. https://doi.org/10.1152/jn.00943.2014

Quek, G.L., Rossion, B., 2017. Category-selective human brain processes elicited in fast periodic visual stimulation streams are immune to temporal predictability. Neuropsychologia 104, 182–200. https://doi.org/10.1016/j.neuropsychologia.2017.08.010

Ray, S., Maunsell, J.H.R., 2011. Different origins of γ rhythm and high-γ activity in macaque visual cortex. PLoS Biol. 9, e1000610. https://doi.org/10.1371/journal.pbio.1000610

Regan, D., 1989. Human Brain Electrophysiology: Evoked Potentials and Evoked Magnetic Fields in Science and Medicine, Elsevier. ed. New York.

Retter, T.L., Jiang, F., Webster, M.A., Rossion, B., 2020. All-or-none face categorization in the human brain. Neuroimage 213, 116685. https://doi.org/10.1016/j.neuroimage.2020.116685

Retter, T.L., Rossion, B., 2016. Uncovering the neural magnitude and spatio-temporal dynamics of natural image categorization in a fast visual stream. Neuropsychologia 91, 9–28. https://doi.org/10.1016/j.neuropsychologia.2016.07.028

Rossion, B., Jacques, C., Jonas, J., 2018. Mapping face categorization in the human ventral occipito-temporal cortex with direct neural intracranial recordings. Ann. N. Y. Acad. Sci. https://doi.org/10.1111/nyas.13596

Rossion, B., Jacques, C., Liu-shuang, J., 2015. Fast periodic presentation of natural images reveals a robust face-selective electrophysiological response in the human brain. J. Vis. 15, (1) 18, 1--18. https://doi.org/10/1167.15.1.18.

Shum, J., Hermes, D., Foster, B.L., Dastjerdi, M., Rangarajan, V., Winawer, J., Miller, K.J., Parvizi, J., 2013. A brain area for visual numerals. J. Neurosci. 33, 6709–15. https://doi.org/10.1523/JNEUROSCI.4558-12.2013

Truccolo, W.A., Ding, M., Knuth, K.H., Nakamura, R., Bressler, S.L., 2002. Trial-to-trial variability of cortical evoked responses: implications for the analysis of functional connectivity. Clin. Neurophysiol. Off. J. Int. Fed. Clin. Neurophysiol. 113, 206–226. https://doi.org/10.1016/s1388-2457(01)00739-8

Wang, Y., Korzeniewska, A., Usami, K., Valenzuela, A., Crone, N.E., 2021. The Dynamics of Language Network Interactions in Lexical Selection: An Intracranial EEG Study. Cereb. Cortex 31, 2058–2070. https://doi.org/10.1093/cercor/bhaa344